# Make Information Diffusion Explainable: LLM-based Causal Framework for Diffusion Prediction

**Wenbo Shang**[*]   **Zihan Feng**[*]   **Yajun Yang**   **Xin Huang**[†]
Hong Kong Baptist University, Tianjin University
{cswbshang, xinhuang}@comp.hkbu.edu.hk, {zihanfeng, yjyang}@tju.edu.cn

## Abstract

Information diffusion prediction, which aims to forecast the future infected users during the information spreading process on social platforms, is a challenging and critical task for public opinion analysis. With the development of social platforms, mass communication has become increasingly widespread. However, most existing methods based on GNNs and sequence models mainly focus on structural and temporal patterns in social networks, suffering from spurious diffusion connections and insufficient information for diffusion analysis. We leverage the strong reasoning capabilities of LLMs and develop an **LL**M-based causal framework for **d**iffusion **in**fluence **d**erivation, named MILD. By comprehensively integrating four key factors of social diffusion—i.e., connections, active timelines, user profiles, and comments—MILD causally infers authentic diffusion links to construct a diffusion influence graph, $G_I$. To validate the quality and reliability of our constructed graph $G_I$, we propose a newly designed set of evaluation metrics for diffusion prediction. In experiments, MILD provides a reliable information diffusion structure that achieves an absolute improvement of 12% over the social network structure and achieves state-of-the-art performance in diffusion prediction. MILD is expected to contribute to higher-quality, more explainable, and more trustworthy public opinion analysis. The code and data are available at: `https://github.com/Shang-hub/MILD-Official-Implementation`.

## 1 Introduction

Information diffusion on social platforms (e.g., TikTok and Instagram) is characterized by a broad reach and intricate, multifaceted pathways, leading to the formation of numerous diffusion cascades. Information diffusion prediction aims to forecast the future participants of cascades, which is crucial for tracing information flow dynamics and quantifying user influence, benefiting downstream applications such as social recommendation systems [21, 8, 15] and public opinion analysis [12, 13, 20]. Therefore, it is fundamentally important to achieve accurate information diffusion prediction, relying on effectively modeling complex user interactions and influences.

Several learning-based approaches have emerged for predicting information diffusion. Initial efforts focus on modeling the sequential nature of diffusion events, primarily employing Recurrent Neural Networks (RNNs) or Transformer architectures to capture temporal influence patterns within observed cascades [11, 31, 32, 33, 35]. More recently, Graph Neural Networks (GNNs) have proven highly effective for graph data, leading to hybrid frameworks that combine GNNs with sequential models to capture both global social network structure and local diffusion dynamics [5, 23, 28, 29, 37, 41]. For instance, methods like MS-HGAT [28] and DisenIDP [3] utilize hypergraph networks to capture user interactions and employ self-attention to model diffusion influence. However, this "GNN+Sequence

---

[*]Equal Contribution.
[†]Corresponding Author.

39th Conference on Neural Information Processing Systems (NeurIPS 2025).

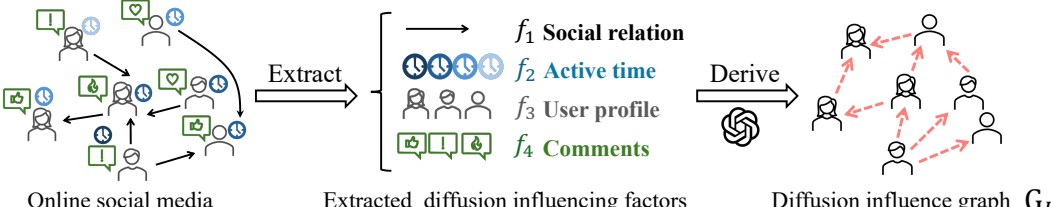

Figure 1: Overview of diffusion influence derivation in the proposed MILD framework.

model" paradigm has limitations in explicitly identifying key influences, which makes it hard to answer a fundamental question: *who influences one to participate in the information diffusion?*

In this paper, we propose a novel model of **LLM**-based **d**iffusion influence **d**erivation (MILD), which aims at tackling the fundamentally key question of who-influences-whom using causal derivation. In particular, MILD contains two derivations: explicitly deriving who-influences-whom diffusion connections in the form of a digraph $G_I$ to represent the cascade, and utilizing the obtained diffusion influence graph $G_I$ to enhance diffusion prediction. We lay the foundation with a key observation: there is an imbalanced diffusion ability for different social actors within cascades. Some are hub influencers, who can activate many inactive actors. But, some are minor influencers, even influencees (i.e., influenced users). For instance, a new participant in a diffusion process is likely to be influenced by close friends or opinion leaders, while unrelated influencees have no diffusion-influencing ability on her/him. However, existing models cannot explicitly capture who-influences-whom connections within a cascade, which blindly takes all participants in a cascade as potential influencers. Furthermore, we have conducted experimental studies to validate this observation in Section 4.3.

For diffusion influence deriving, as the authentic diffusion process is *unknown*, *unexplainable*, and *non-transparent*, it becomes challenging. Leveraging strong reasoning capabilities, MILD employs the LLM like a sociology expert at a high level to comprehensively analyze four key diffusion influencing factors to derive who-influences-whom pairs, as shown in Figure 1. MILD extracts four key factors of social relationships, active timestamps, user profiles, and comments. Integrating these key factors, MILD finally derives a diffusion influence graph $G_I$ to represent the cascade diffusion. To discover effective pathways for $G_I$ construction, we relax the formulated NP-hard problem and develop fast heuristic strategies to find potential diffusion pathways in polynomial time.

Additionally, we propose a well-designed set of evaluation metrics to verify the reliability of diffusion influence graph $G_I$, which aims at capturing real key diffusion pathways. To reveal that social relationship significantly differs from the information diffusion, we conduct a comparison analysis visually and quantitatively to show the differences between our $G_I$ and social relationships in Section 3.3. We found that $G_I$ is more reliable and effective, representing 12% more real-world diffusion pathways and 28% more key interactions compared with the social network.

Finally, we integrate the diffusion influence graph $G_I$ to enhance the performance on the diffusion prediction task and empirically demonstrate the effectiveness of MILD on two real-world datasets with diffusion benchmarks. Extensive experiments validate the superiority of MILD against eight state-of-the-art competitors [37, 28, 3, 23, 14, 5, 29, 41], by achieving ~7% improvement on average.

## 2 Preliminaries

The social network $G = (V, E, O)$ is a directed graph, where $V$ is the set of users, $E \subseteq V \times V$ is the set of directed social relations (e.g., follower–followee), and $O = \{o_i \mid v_i \in V\}$ is the set of user profiles, with each $o_i$ contains attributes such as follower count, bi-follower count, and status count, etc. An information diffusion cascade on social network $G$ is an ordered sequence $C = ((v_1, t_1, b_1), \ldots, (v_k, t_k, b_k))$, where $v_i \in V$ and $v_i \neq v_j$ for all $i \neq j$; $t_1 \leq \cdots \leq t_k$ are the absolute timestamps at which users become active; $b_i$ denotes the behavior of user $v_i$ (e.g., like, comment, share). $|C|$ denotes the cascade length and $|\mathcal{C}|$ denotes the total number of cascades. Note that throughout this paper, all mentioned paths are simple paths without loops.

Given a cascade $C$ and the social network $G$, the objective of MILD is to derive the diffusion influence graph $G_I$ to represent the diffusion process based on four extracted key factors $\{\psi, T, O, B\}$ to

enhance performances on *diffusion prediction*, where $\psi$ denotes potential diffusion paths derived from social networks, $T$ denotes the temporal order of participants based on their active time, $O$ denotes the influence and activity levels of users derived from profiles, and $B$ denotes the comments.

**Diffusion influence graph**. The diffusion influence graph is a digraph $G_I = (V_I, E_I)$, where $V_I \subseteq V$ is the set of cascade participants ($|V_I| = |C| = k$), and $E_I$ represents potential influence relationships. We represent the (binary) influence matrix of $G_I$ as $\mathbf{A}^I \in \{0,1\}^{k \times k}$, where $\mathbf{A}^I_{ij} = 1$ represent $v_i$ influences $v_j$, i.e., $(v_i \to v_j) \in E_I$, otherwise 0.

**Problem formulation of diffusion prediction.** Given an observed cascade $C$ with $k \in \mathbb{Z}^+$ existing participants, and an influence matrix $\mathbf{A}^I$, the problem of diffusion prediction is to find $\gamma \in \mathbb{Z}^+$ users as the candidate participants $\text{Top}_\gamma(V)$ in future. Specifically, $\text{Top}_\gamma(V) = \{v \in V : f_\theta(v \mid \mathbf{A}^I) \geq f_\theta(u \mid \mathbf{A}^I), \forall u \in V \setminus \text{Top}_\gamma(V)\}$ and $|\text{Top}_\gamma(V)| = \gamma$, where $f_\theta(\cdot)$ is a learned scoring function over all users in the social network $G$ and $\theta$ denotes the parameter of the learning model.

# 3 Methodology

This section introduces a causal framework to enhance accurate diffusion prediction. Section 3.1 introduces a causal view $C \to Z \to Y$, emphasizing the necessity of explicit key influence representation $Z$. Section 3.2 outlines the method for prompting LLMs to infer a *who-influences-whom* graph, converting each cascade $C$ into a structured diffusion influence digraph $Z$. Section 3.3 proposes three evaluation metrics and a comparison analysis to assess the reliability and consistency of the derived $Z$. Section 3.4 integrates the LLM-derived diffusion influence $Z$ into a downstream predictor $Y$, explicitly highlighting key influences to enhance prediction in practice.

## 3.1 Causal View on Information Diffusion

To better understand the limitations of existing information diffusion prediction methods, we take a causal view of $P(Y|C)$ and construct a structural causal model (SCM) [22] in Figure 2(a). It describes the causalities among three variables, i.e., the information diffusion cascade $C$, spurious and incomplete diffusion pathways $U$, and the diffusion prediction label $Y$. Specifically: (1) $C \to Y$ describes that existing works predict the next participant $Y$ by learning from the observed historical cascade $C$; (2) $C \leftarrow U \to Y$ shows the dependence between $C$ and $Y$ induced by the spurious and incomplete diffusion process $U$. To formulate the true causal relationship between $C$ and $Y$, we employ the back-door adjustment to estimate $P(Y|do(C))$ as follows. The detailed analysis is in Appendix C.1.

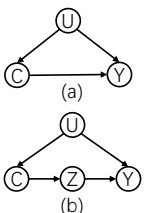

Figure 2: SCM.

$$P(Y|do(C)) = \sum_{u \in U} P(U = u)P(Y|C = c, U = u), \tag{1}$$

where $u$ denotes a specific spurious diffusion process of $U$. In Eq. 1, we observe that $P(Y|do(C))$ comprises spurious diffusion directions, which may lead to incorrect decisions.

In this paper, we add an intermediate variable $Z$ between $C$ and $Y$, as the representation of the diffusion influence shown in Figure 2(b), satisfying the Front-door Adjustment. In our SCM, $C \to Z \to Y$ shows that the diffusion influence $Z$ is derived from the cascade $C$. Then, MILD employs $Z$ as the explicit diffusion cascade representation to make an accurate diffusion prediction label $Y$. Based on the detailed analysis in Appendix C.2, we can formulate our model by chaining together the two partial effects and summing over all states $z$ of $Z$ as follows:

$$P(Y|do(C)) = \sum_{z \in Z} \sum_{c \in C'} P(Y|Z, C')P(C')P(Z|C), \tag{2}$$

where $C'$ contains all observed cascade sequences in the training data, and $z$ denotes a specific diffusion influence matrix in $Z$. In Eq. 2, we can observe that when MILD captures the diffusion influence as closely as possible to the real-world one, we can achieve accurate prediction results. Therefore, MILD contains two parts: deriving explicit diffusion influence from the cascade ($C \to Z$), and making diffusion prediction based on the derived diffusion influence ($Z \to Y$).

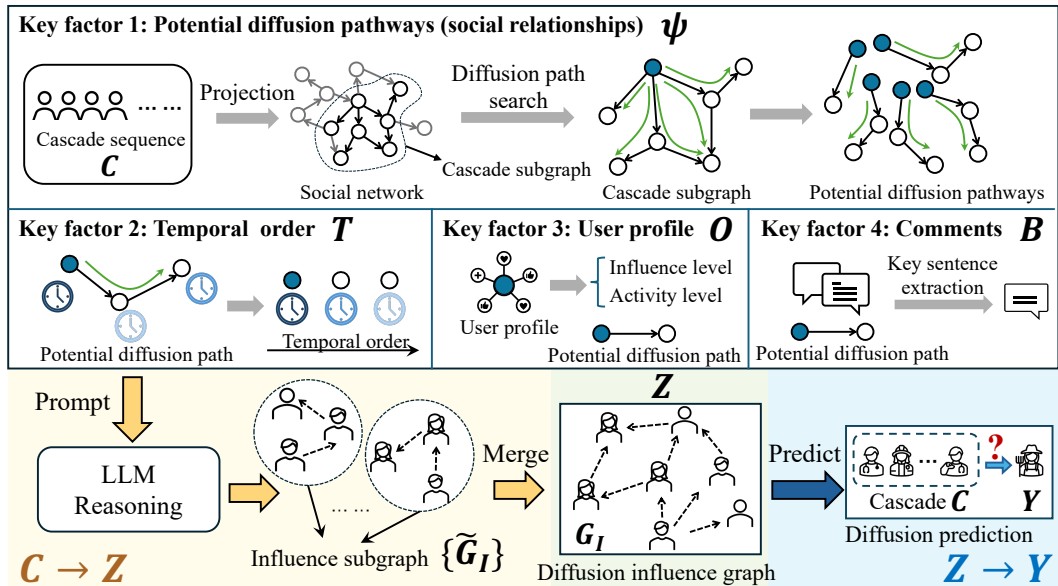

Figure 3: The proposed MILD framework. (1) $C \rightarrow Z$: To capture $Z$, we construct four essential diffusion influencing factors for the LLM to understand and infer the diffusion process, which are social relationships $\psi$, temporal order $T$, user profiles $O$, and shared comments $B$. LLM-derived diffusion influence subgraphs are merged by aligning shared nodes and edges to form the final diffusion influence graph $Z$. (2) $Z \rightarrow Y$: A crafted cascade module integrates the structured diffusion influence graph $G_I$ into the predictor, which can explicitly highlight key influences.

## 3.2 Diffusion Influence Deriving: $C \rightarrow Z$

In this part, we introduce our MILD to explicitly identify and extract the diffusion process of cascade $C$ into a structured diffusion influence graph $G_I$ as the intermediate variable $Z$, making the diffusion process transparent and explainable, as shown in Figure 3.

Given the current limitations of LLMs in reasoning about and structurally comprehending digraphs [26, 27, 10, 19, 30, 39], we propose constructing potential diffusion paths to represent social relationships, thereby enhancing the LLM's understanding of social diffusion patterns. Specifically, for a given cascade sequence $C$ and social network $G$, we first extract the induced subgraph consisting of all participants in $C$, forming the cascade graph $G_C = (V_C, E_C)$. Here, $V_C$ represents the set of cascade participants, and $E_C$ represents their social links within $G$. Subsequently, we transform the cascade graph $G_C$ into a set of potential diffusion paths, denoted as $\Psi$. We define this process as the Diffusion Path Search problem. Its objective is to obtain a diverse and high-quality set of paths $\Psi$ suitable for LLM analysis.

**Definition 1: [Diffusion path search problem]** Given a cascade graph $G_C = (V_C, E_C)$ with one unique root $v_1$, the problem returns a collection of path structures $\Psi \subseteq G_C$ by satisfying the following three constraints: (D1) all paths $\psi \in \Psi$ start from the unique root $v_1$; (D2) $V_\Psi$ covers all nodes in $V_C$; and (D3) $\Psi$ contains all possible paths between $v_1$ and $v \in \{v_i | dist(v_1, v_i) \geq \ell\}$.

**Theorem 1:** *The diffusion path search problem is NP-hard.*

The detailed proof of Theorem 1 is shown in Appendix C. Given the NP-hardness of this problem, we propose a heuristic alternative algorithm based on the combination of Breadth First Search (BFS) and Depth First Search (DFS). This algorithm balances *diffusion consistency* (long, deep paths) against *diffusion coverage* (short, shallow paths). First, we conduct the single-source BFS algorithm with the original participant $v_1$ as the single source in $O(|V_C| + |E_C|)$ time. The algorithm returns the BFS tree $T_{\text{BFS}}$ composed of shortest paths between $v_1$ and $v \in V_C$, satisfying (D1) and (D2). In $T_{\text{BFS}}$, longer shortest paths signify greater social distance between participants, typically involving more complex information diffusion routes. In contrast, shorter paths indicate closer social distance

and simpler diffusion routes. Based on this analysis, for each long shortest path in $T_{\text{BFS}}$, we select its source $s$ and target $\tau$, and then search all simple paths between $s$ and $\tau$ in $G_C$ using the DFS algorithm, satisfying (D3). However, in the complete graph of order $n$, all simple paths can be found in $O(n!)$ time. Considering efficiency, we design a strict constraint on the density of $G_C$ (details in Appendix B.3). If the density is less than $r$, we search all simple paths between $s$ and $\tau$ in $O(|V_C|d_{max}^r)$ time, where $d_{max}$ denotes the maximum out-degree in $G_C$. Otherwise, we search all shortest paths between $s$ and $\tau$ in $O(|V_C||E_C|)$ time. We iterate this process for each long shortest path in $T_{\text{BFS}}$. Meanwhile, we preserve short paths in $T_{\text{BFS}}$ that exclude the target from long paths for (D2). Since $O(|V_C| + |E_C|) \subseteq O(|V_C|d_{max}^r + |V_C||E_C|)$, the total time complexity is $O(|V_C|d_{max}^r + |V_C||E_C|)$. Therefore, the algorithm can effectively search diffusion paths that satisfy (D1), (D2), and (D3) in polynomial time $O(|V_C|d_{max}^r + |V_C||E_C|)$.

Building on the preceding analysis, we design four critical factors for analyzing diffusion influence. First, the potential diffusion pathway set $\Psi$ represents social relationships. Second, the temporal order $T$, which tracks the time of participant comments or shares, reflects the order of their involvement. Third, user influence and activity levels $O$, which are estimated under the guidance of communication theories from user profiles [2, 7, 4]. The detailed calculation process can be found in Appendix B.2. We assign a specific level for each estimation, i.e., 0–20%: poor, 20%–40%: low, 40%–80%: moderate, 80%–95%: high, 95%–100%: outstanding. Lastly, the comment contents shared by participants $B$. We perform key sentence extraction on $B$ based on the TextRank algorithm [16] to mitigate the limitations of input length in language models and enhance key semantics in $B$.

**Diffusion influence deriving.** We design the prompt to instruct the LLM to analyze the diffusion influence between users within each cascade. The potential diffusion path $\psi \in \Psi$, temporal order $T$, descriptions of user influence and activity levels $O$, and comments $B$ are integrated into the designed template $\Phi(\cdot)$, forming prompts $\Phi(\psi, T, O, B)$ to provide sufficient information for analysis. To guarantee the reliability of the derivation, we add basic principles of information diffusion to the design of the instruction. For instance, later participants do not influence earlier participants. The detailed prompt can be found in Appendix Figure 14. Then, we feed instructions and prompts into the LLM to derive diffusion influence in the form of user-pair lists denoted as $[[v_i, v_j], ...]$, where $v_i$ conveys the information to $v_j$ and influences $v_j$ to participate in this cascade.

**Out-of-the-box diffusion influence representation.** For each potential diffusion path, the LLM infers critical diffusion influences within paths, namely $[[v_i, v_j], ...]$, which form a directed influence subgraph $\tilde{G}_I$. As each cascade contains multiple diffusion paths, a single cascade corresponds to a set of several influence subgraphs $\{\tilde{G}_I\}$. We then merge $\{\tilde{G}_I\}$ into a unified diffusion influence graph $G_I$ by aligning shared nodes and edges across these subgraphs, as shown in Figure 3. In $G_I$, each node represents one participant in the corresponding cascade, and each directed edge $\langle v_i, v_j \rangle$ signifies that user $v_j$ engages in the cascade under the influence of user $v_i$. Thus, $G_I$ represents the *transparent* and *explainable* diffusion process and functions as the intermediate variable $Z$.

### 3.3 Measurements of Diffusion Influence $G_I$

To evaluate the reliability and consistency of $G_I$ derived from MILD, we carefully design quantitative measures: (E1) **Diffusion pathways evaluation** assesses whether $G_I$ can accurately capture the real-world information diffusion routes. (E2) **Key interaction evaluation** examines whether $G_I$ observes participant interactions within cascades. (E3) **Robustness evaluation** assesses whether MILD can distinguish real and fake diffusion pathways under the interference of misleading instructions, ensuring consistent and reliable responses. (E4) **Comparison of $G_I$ and $G_C$** analyzes the role of social relationships in the diffusion process. Our findings are as follows:

1. **The social network fails to effectively capture the real-world diffusion process** compared to our $G_I$. Although it can establish the necessary social relationships to represent partial diffusion connections, it can scarcely capture key and implicit diffusion connections. (E1&E2)
2. **Multiple diverse diffusion pathways can enhance MILD's inference ability** in diffusion influence derivation compared to shortest relationships, indicating that the diffusion process is complex and cannot solely depend on the closest social relationships. (E1&E2)
3. **Temporal order is critical** for analyzing the diffusion process. (E1&E2)
4. **MILD can distinguish real and fake diffusion processes under the interference of misleading instructions** and generate consistent results of diffusion influence analysis. (E3)

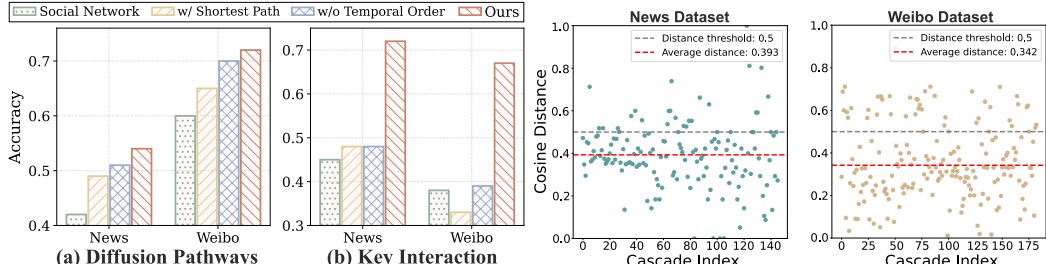

Figure 4: Reliability evaluation of $G_I$.  Figure 5: Comparison of $G_I$ and $G_C$.

5. **The forming process of social relationships differs from the diffusion process**, although social relationships are significant for MILD in analyzing the diffusion process. (E4)

**(E1) Diffusion pathways evaluation.** In real-world data, diffusion pathways often appear in reply chains, in the form of $\{A : [\text{comment}]@B : [\text{comment}]@C : [\text{comment}]\}$, where @ indicates the reply and share behavior during the diffusion process. Thus, we can extract these real-world diffusion pathways as the ground truth, such as C conveying the information to B, who then passed it on to A, formulated as [[C, B], [B, A]]. Then, we evaluate $G_I$ based on our extracted ground-truth $G_{gt}$. Considering the scarcity and absence of reply comments in the dataset, we design the following evaluation method: (1) for each cascade, we evaluate $G_I$ and $G_{gt}$ once; (2) calculate the number of intersecting directed edges between $G_I$ and $G_{gt}$; (3) calculate the proportion of the intersection set in $G_{gt}$ and then average the overall results, which can be formulated as follows.

$$\text{Accuracy} = \frac{1}{|\mathcal{G}_{gt}|} \sum_{G_{gt} \in \mathcal{G}_{gt}} \frac{|E(G_i) \cap E(G_{gt})|}{|E(G_{gt})|}, \tag{3}$$

where $E(\cdot)$ denotes the edge set of the given graph and $|\mathcal{G}_{gt}|$ denotes the number of $G_{gt}$ in the total set $\mathcal{G}_{gt}$. To verify the effectiveness of our method, we compare it with three representations of diffusion influence: (1) the original social network, (2) our method with the shortest potential diffusion paths, and (3) our method without temporal order. As shown in Figure 5(a), the social network fails to accurately capture the real-world diffusion process. Furthermore, diverse potential diffusion pathways and temporal order are critical and can provide necessary information for MILD to analyze the diffusion process.

**(E2) Key interaction evaluation.** As the well-known communication theory, the Strength of Weak Ties [7] posits that *"the strength of any tie is a function of the frequency and duration of interaction,"* we assess whether $G_I$ can observe the key interactions among participants. We regard high-frequency interactions as key interactions. To ensure fairness in evaluation and avoid information leakage, we split the dataset by the cascade initiation time in an 8:2 ratio. We assess whether our method can observe key interactions in the 20% of newly occurred cascades based on the interactions in the 80% of historical cascades. The evaluation metric and baselines are the same as in the diffusion pathways evaluation. As shown in Figure 5(b), compared to the other representations, we find that $G_I$ more effectively captures strong ties and key interactions among participants in new cascades.

**(E3) Robustness evaluation.** To evaluate whether MILD can distinguish real and fake diffusion paths, we construct pairs of such paths and intentionally mislead the LLM to assume that all paths are real to analyze the influence strength. Initially, for each cascade, we generate multiple fake diffusion paths by retaining the same source and target nodes as the real paths, but randomly replacing the intermediate nodes with unrelated participants. We then mix all paths and prompt MILD to analyze the pair-wise influence strength within these "real" diffusion paths as $P(v_i|v_{i-1}) \in [0, 1]$, where $v_{i-1}$ is the predecessor of $v_i$ in the path $\psi$. We calculate the path influence strength $P(\psi)$ and the margin difference between real and fake diffusion paths. The path influence strength of the true path $\psi$ is denoted as $P_{true}(\psi)$. Similarly, the influence strength of a fake path $\psi'$ is $P_{fake}(\psi')$.

$$P(\psi) = \prod_{i=2}^{n} P(v_i|v_{i-1}, ..., v_1)P(v_1) = \prod_{i=2}^{n} P(v_i|v_{i-1})P(v_1), \tag{4}$$

$$\text{margin}_C = \frac{1}{|\Psi_{real}|} \sum_{\psi \in \Psi_{real}} P_{true}(\psi) - \frac{1}{|\Psi_{fake}|} \sum_{\psi' \in \Psi_{fake}} P_{fake}(\psi'), \quad \Psi_{real}, \Psi_{fake} \subseteq C, \quad (5)$$

Based on the calculation, we introduce the Correct Distinguishing Rate (CDR), which assesses instances where the margin difference exceeds a specified threshold $\delta$.

$$f(\text{margin}_C) = \begin{cases} 1 & \text{if } \text{margin}_C > \delta \\ 0 & \text{if } \text{margin}_C \le \delta \end{cases}, \quad \text{CDR} = \frac{\sum\limits_{C \in \mathcal{C}} f(\text{margin}_C)}{|\mathcal{C}|}. \quad (6)$$

Detailed proof is provided in Appendix C.8. We adjust the threshold $\delta$ of $\text{margin}_{C_i}$ from 2% to 14% to observe MILD's ability to distinguish, as depicted in Figure 6. Our findings indicate that MILD can effectively distinguish real and fake diffusion paths with a margin, even under the interference of misleading instructions, showing strong robustness. The results consistently show that real paths have a greater diffusion influence strength than fake paths. Even with a $\delta$ value of 10%, the CDR remains around 0.8 across two datasets. The detailed prompt is in Figure 16.

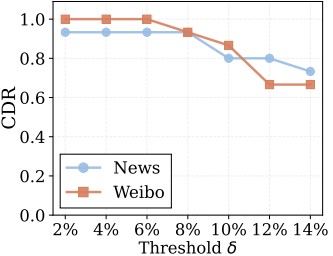

Figure 6: Robustness evaluation.

**(E4) Comparison of $G_I$ and $G_C$.** We quantify the differences between our diffusion influence graph $G_I$ and the social relationship graph $G_C$, analyzing the role of social relationships in the diffusion process. For each cascade, we calculate the cosine distance as the difference between $G_I$ and $G_C$. $\mathbf{A}^I \in \mathbb{R}^{k \times k}$ and $\mathbf{A}^C \in \mathbb{R}^{k \times k}$ are the adjacency matrices of $G_I$ and $G_C$, respectively, where $k$ denotes the number of participants in the cascade. The detailed calculation process is below.

$$\mathbf{v_I} = \mathbf{flatten}(\mathbf{A}^I) \in \mathbb{R}^{k \times k}, \quad \mathbf{v_C} = \mathbf{flatten}(\mathbf{A}^C) \in \mathbb{R}^{k \times k}, \quad (7)$$

$$\mathbf{dis}(\mathbf{A}^I, \mathbf{A}^C) = 1 - \frac{\mathbf{v_I} \cdot \mathbf{v_C}}{\|\mathbf{v_I}\| \cdot \|\mathbf{v_C}\|}. \quad (8)$$

We calculate the matrix distances of all cascades in Figure 5. The range of $\mathbf{dis}(\mathbf{A}^I, \mathbf{A}^C)$ is [0, 1]. The gray line separates close- and far-distance samples. The red line represents the average distance between $G_I$ and $G_C$. The results indicate that $G_I$ and $G_C$ are relevant, as more samples fall in the close area. However, most $G_I$ structures differ from those of $G_C$, with a 0.35 distance on average. Our findings suggest that although social relationships are a key factor in the diffusion process, the way these relationships are structured differs from the diffusion process itself.

### 3.4 Diffusion Prediction: $Z \to Y$

To empirically verify the effectiveness, we further evaluate $G_I$ via the downstream prediction task. Following common workflows [23, 28, 29, 41], we use a two-phase paradigm. First, following [6], we employ the subgraph-centric encoder to learn the social network $G$. Second, a crafted attention module captures the diffusion cascades. Specifically, we aim to harness an LLM-derived *diffusion influence graph* $G_I$ as an explicit structured intermediate representation $Z$ for each cascade $C$. A key design principle is the *controlled exclusion* of raw contextual features (e.g., user profile attributes or full-text embeddings) from the downstream predictor. This ensures that *any gains in prediction accuracy can be attributed solely to the structured diffusion influence information encoded within $G_I$*, rather than to auxiliary side information. Formally, we inject the adjacency matrix $\mathbf{A}^I$ of $G_I$ directly into the self-attention computation. Given participant embeddings $\mathbf{H}_k \in \mathbb{R}^{k \times d}$, we calculate

$$\mathbf{Q} = \mathbf{H}_k \mathbf{W}^Q, \quad \mathbf{K} = \mathbf{H}_k \mathbf{W}^K, \quad \mathbf{V} = \mathbf{H}_k \mathbf{W}^V, \quad (9)$$

$$\text{Attn}(\mathbf{Q}, \mathbf{K}, \mathbf{V}) = \text{softmax}\left( \frac{\mathbf{Q} \mathbf{K}^\top}{\sqrt{d'}} + \mathbf{M} + \boxed{\alpha \mathbf{A}^I} \right) \mathbf{V}. \quad (10)$$

where $\mathbf{W}^Q, \mathbf{W}^K, \mathbf{W}^V \in \mathbb{R}^{d \times d}$ are learnable projections, $d' := d/B$ with $B$ attention heads, and $\mathbf{M} \in \{0, -\infty\}^{k \times k}$ enforces causality by masking future positions. The scalar $\alpha$ is a temperature factor that calibrates the influence graph's impact on attention scores. Subsequently, the multi-head outputs are concatenated and processed by a position-wise feed-forward network to yield the final cascade representation $\mathbf{H}_c$, which is distilled solely from $G_I$ for downstream prediction tasks.

Table 2: Performance of information diffusion prediction (averaged from 5 runs). Higher scores are better. Corresponding standard deviations (±) are provided in the Appendix A.3.

| Datasets | News | | | | | | Weibo | | | | | |
|---|---|---|---|---|---|---|---|---|---|---|---|---|
| Metrics | H@10 | H@50 | H@100 | M@10 | M@50 | M@100 | H@10 | H@50 | H@100 | M@10 | M@50 | M@100 |
| DyHGCN [37] | 18.94 | 24.50 | 27.31 | 10.50 | 10.76 | 10.80 | 10.51 | 15.39 | 18.50 | 6.01 | 6.23 | 6.27 |
| MSHGAT [28] | 20.10 | 25.82 | 28.85 | 11.68 | 11.95 | 11.99 | 11.41 | 18.34 | 21.77 | 6.16 | 6.48 | 6.53 |
| DisenIDP [3] | 20.47 | 26.33 | 29.08 | 12.03 | 12.29 | 12.36 | 12.04 | 18.83 | 22.61 | 6.73 | 7.01 | 7.06 |
| RotDiff [23] | 20.95 | 27.02 | 29.80 | 12.52 | 12.80 | 12.84 | 12.99 | 20.46 | 24.70 | 7.76 | 8.12 | 8.18 |
| MINDS [14] | 20.04 | 25.66 | 28.70 | 11.72 | 11.93 | 11.98 | 11.69 | 18.07 | 21.66 | 6.24 | 6.51 | 6.57 |
| MGCL [5] | 21.26 | 28.11 | 31.93 | 12.90 | 13.18 | 13.24 | 13.04 | 20.18 | 24.75 | 7.60 | 7.99 | 8.03 |
| GODEN [29] | 22.01 | 29.14 | 33.17 | 13.34 | 13.67 | 13.73 | 14.08 | 21.95 | 26.71 | 8.09 | 8.28 | 8.52 |
| CARE [41] | 22.47 | 28.71 | 32.01 | 14.53 | 14.81 | 14.85 | 13.35 | 21.96 | 26.62 | 7.65 | 8.05 | 8.11 |
| **MILD (Ours)** | **24.07** | **31.33** | **35.52** | **15.50** | **15.85** | **15.91** | **14.71** | **23.37** | **28.17** | **8.71** | **9.10** | **9.17** |
| *Improvement (%)* | +7.12 | +7.52 | +7.08 | +6.67 | +5.19 | +7.14 | +4.47 | +6.42 | +5.47 | +7.66 | +9.89 | +7.63 |

# 4 Experiments

**Datasets.** We mainly evaluate the performance on two real-world datasets, *News* and *Weibo*, including the social network structure, detailed user profiles, and information diffusion cascades. In particular, a cascade contains participant IDs with specific participation behavior, timestamps, associated comments, and the original post. Following general settings, we split the cascades in an 8:1:1 ratio for training, validation, and testing in the order of post time. Additionally, we further conduct empirical analysis to support our motivations on four widely used datasets: *Twitter* [9], *Douban* [40], *Android* [24], and *Christianity* [24]. Owing to the absence of textual data (participation behavior, user profiles, the original post, and comments) for LLM reasoning, we only use them for motivation analysis in Appendix A.2.

Table 1: Statistics of the used datasets.

| Datasets | News | Weibo |
|---|---|---|
| # Users | 10,255 | 31,061 |
| # Edges | 83,959 | 294,577 |
| # Cascades | 1,291 | 1,910 |
| Avg Len. | 63.45 | 175.46 |

**Competitors.** We evaluate the proposed *MILD* against eight state-of-the-art models for information diffusion prediction: *DyHGCN* [37], *MS-HGAT* [28], *DisenIDP* [3], *RotDiff* [23], *MINDS* [14], *MGCL* [5], *GODEN* [29], and *CARE* [41]. Generally, these methods adopt a two-phase framework—"Graph + Cascade"—that employs GNNs to learn various user relations, and then uses RNNs or Self-Attention to learn sequential cascades. Following standard settings [5, 29, 41], we use *Hit rate* (Hits@K) and *Mean Average Precision* (MAP@K) to evaluate the performance of information diffusion prediction. For comprehensive evaluation, we report the results at $K \in \{10, 50, 100\}$. We provide more details of competitors and metrics in Appendix D.2 and D.3.

**Implementation Details.** In the diffusion path search algorithm, we impose the graph-density constraint $r = 1.5$ for efficiency. For LLM inference, we use GPT-4o with the temperature set to 0 and top_k set to 1 to ensure the reproducibility of $G_I$, leaving all other parameters at their default values. For the diffusion predictor, model weights are optimized with AdamW at a learning rate of 1e-3. The number of GAT layers is 1, and the number of attention heads is 6. $\alpha$ defaults to 1. Some hyperparameters are analyzed in Appendix A.1. We train with mini-batches of size 32, using 256-dimensional embeddings, for up to 50 epochs. All baselines are re-evaluated under their original published settings, with a uniform maximum cascade length of 200 for fair comparison. All experiments are conducted on 2 NVIDIA RTX V100 (32 GB) GPUs.

## 4.1 Diffusion Prediction

Table 2 demonstrates that our MILD significantly outperforms eight state-of-the-art baselines on two real-world information diffusion prediction datasets, achieving average improvements of 6.3% on Hits@K and 7.4% on MAP@K. Unlike leading methods such as CARE and GODEN, which model social network structure and temporal sequences to implicitly capture patterns, **MILD's core distinction is the explicit modeling and integration of a structured diffusion influence graph derived using LLMs.** This directly represents the underlying dynamics driving user participation, a factor often lacking explicit capture in prior work. These improvements underscore two key insights: (1) LLMs can discern causal influence patterns beyond surface-level statistics; and (2) integrating this influence structure enhances the cascade learner's power by focusing on critical transmission links. Crucially, these gains are achieved **without** using additional raw contextual data as direct input

Table 3: Ablation studies on information diffusion prediction.

| Datasets | News | | | | | | Weibo | | | | | |
|---|---|---|---|---|---|---|---|---|---|---|---|---|
| Metrics | H@10 | H@50 | H@100 | M@10 | M@50 | M@100 | H@10 | H@50 | H@100 | M@10 | M@50 | M@100 |
| **w/o** Path Selection | 23.14 | 29.20 | 33.72 | 14.53 | 14.76 | 14.82 | 14.33 | 21.96 | 27.15 | 8.44 | 8.82 | 8.87 |
| **w/o** Post Content | 23.72 | 29.98 | 33.65 | 14.90 | 15.21 | 15.24 | 14.58 | 22.87 | 27.64 | 8.51 | 8.96 | 9.02 |
| **w/o** Time Information | 23.16 | 30.08 | 33.97 | 14.77 | 14.83 | 14.89 | 14.36 | 22.15 | 26.83 | 8.49 | 8.84 | 8.89 |
| **w/o** User Profile | 23.77 | 30.73 | 34.62 | 14.96 | 15.24 | 15.28 | 14.52 | 22.81 | 27.46 | 8.52 | 8.90 | 8.93 |
| **w** Social bias | 22.92 | 27.64 | 31.35 | 14.68 | 14.75 | 14.90 | 14.33 | 22.12 | 26.64 | 8.45 | 8.77 | 8.96 |
| **w/o** Influence bias | 21.81 | 27.52 | 30.62 | 14.03 | 14.31 | 14.35 | 14.08 | 21.98 | 26.21 | 8.26 | 8.62 | 8.68 |
| **Full Model** | **24.07** | **31.33** | **35.52** | **15.50** | **15.85** | **15.91** | **14.71** | **23.37** | **28.17** | **8.71** | **9.10** | **9.17** |

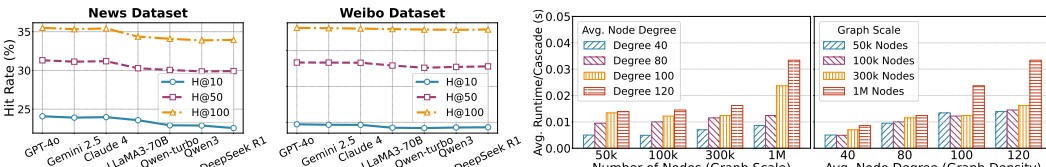

Figure 7: Stability across different LLMs.   Figure 8: Runtime of our heuristic algorithm.

for the prediction task. This controlled setting rigorously validates that the predictive power stems directly from the *structured diffusion influence representations* extracted by our LLM. These findings highlight the value of explicit influence modeling in diffusion and the insights of LLMs.

## 4.2 Ablation Study

To investigate the contributions of MILD's components, particularly LLM-based influence identification and its subsequent integration, we conduct comprehensive ablation studies in Table 3. **(1) Ablation on Four Key Factors:** We first ablate the four key factors used by the LLM to identify the diffusion influence graph. Removing any single factor—*w/o Structure*, *w/o Post Content*, *w/o Time Information*, or *w/o User Profile*—consistently degrades performance, which verifies their necessity in the diffusion influence analysis. The largest performance drop, seen in *w/o Structure*, highlights the significance of our novel algorithm of potential diffusion pathways. **(2) Ablation on $G_I$:** We evaluate whether $G_I$ effectively enhances diffusion prediction performance and more accurately represents the diffusion process than social networks. The *w/o Influence bias* variant (removing influence graph integration) results in a significant performance drop, validating the necessity of explicit influence modeling $G_I$. The *w Social bias* variant (using simple social adjacency instead of our $G_I$) shows only marginal improvement over *w/o Influence bias*. This significant gap between *w Social bias* and MILD convincingly proves that $G_I$ can more accurately represent the diffusion process than social networks, and truly capture the nuanced influence relationships to enhance prediction performance.

**MILD across Different LLMs.** To evaluate the adaptability of MILD across a diverse range of LLMs, we measure performance on six advanced models: Gemini 2.5, Claude 4, LLaMA 3, Qwen 3, Qwen-Turbo, and DeepSeek R1. As illustrated in Figure 7, MILD exhibits consistently strong performance across all variants, indicating stable behavior under changes in the underlying LLM.

**Efficiency Evaluation of our Heuristic Algorithm.** We evaluate the computational efficiency of our heuristic algorithm on a range of graphs with varying scales and densities. As shown in Figure 8, our heuristic solution demonstrates strong efficiency, achieving a time cost of 0.0334 seconds per cascade, even on large-scale, dense graphs comprising millions of nodes and an average degree of 120.

## 4.3 Empirical Analysis on Influence

**Influence Intervention.** To validate the existence of spurious influences in cascades, we conduct Influence Intervention experiments. Self-attention mechanisms are widely adopted to capture inter-node influence [28, 29, 37]. We randomly mask $k\%$ of the self-attention weights (setting the masked Q_K values to $-\infty$) and observe the results. Interestingly, Figure 9 shows that randomly masking some influences does not critically impair performance—it even performs better than full attention when less than 15% are masked, indicating the presence of numerous spurious diffusion influences. Therefore, accurately identifying key influences is crucial for enhancing prediction performance. This observation is also evident on the widely used benchmark datasets (see Appendix A.2).

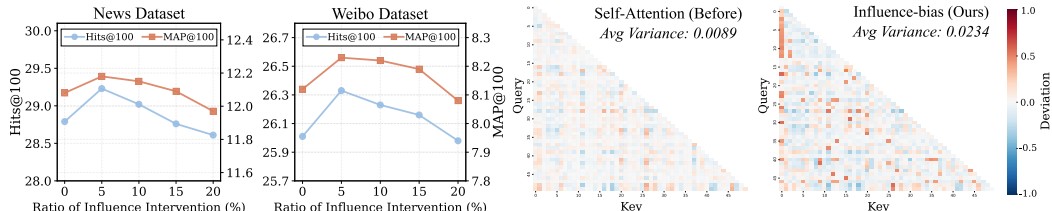

Figure 9: Influence intervention.      Figure 10: Attention visualization.

**Attention Visualization.** To better understand the role of $G_I$, we further visualize the Q_K attention matrix for a case from the Weibo dataset (Appendix A.2 provides more cases). As shown in Figure 10, the left shows self-attention, and the right shows our MILD with influence bias. Furthermore, we also compute the variance of attention weights to provide a quantitative assessment. Both the visual patterns and the increased variance indicate that our proposed MILD effectively amplifies key diffusion influences, leading to the significant performance gains in diffusion prediction.

# 5 Related Work

Information diffusion on social media reflects how users influence one another and propagate opinions across a network. Information Diffusion Prediction aims to identify which users will be "infected" by a piece of content, given observed diffusion cascades and the social graph [34, 8, 18, 13]. Recent approaches typically combine GNNs with sequence models to capture, respectively, the social network structure and the temporal ordering within cascades [36, 37, 28, 23, 3, 5, 29, 41, 25]. Concretely, GNNs are often used to learn homophilous ties—reflecting that friends often share interests—while sequence models, such as RNNs or self-attention, encode the dependencies among users within a specific cascade. For example, GODEN [29] employs an ODE-based GNN to track dynamic user relations before applying self-attention to model the cascade sequence. CARE [41] uses a GAT to embed the social graph, and then applies a Transformer for cascade modeling enhanced by retrieved in-context historical cascades to enrich current predictions. However, despite their effectiveness in modeling structure and sequence, these methods often face significant challenges in explicitly distinguishing the nuanced influence mechanisms driving user participation in a diffusion process. Furthermore, a common constraint across existing works is relying solely on network structure and basic cascade metadata (user IDs and timestamps). They largely overlook richer contextual signals, such as detailed user profiles, post content, and interaction metadata. In this work, we introduce two new datasets featuring such rich contextual information. Leveraging these datasets, we propose MILD, a framework that explicitly exploits and understands the diverse diffusion influences, enabling more explainable prediction and significant performance improvements.

# 6 Conclusion and Future Work

We propose MILD, a novel diffusion influence inference framework that builds on four diffusion influencing factors and guides LLMs to comprehend and infer the authentic diffusion process, thereby improving diffusion prediction performance. We show that MILD provides a reliable diffusion process analysis that achieves 12% absolute improvement over the social network structures. We conduct extensive experiments on large-scale real-world datasets and verify the effectiveness of MILD. This diffusion influence analysis further contributes to better and more explainable public opinion analysis.

**Limitations.** This work is limited by the availability of accessible real-world datasets composed of user profiles, social network structure, and diffusion dynamics. Due to privacy concerns and government regulations, accessing extensive social and public opinion data has become increasingly challenging. Recognizing the necessity of such datasets, we are dedicated to addressing this limitation by legally constructing novel benchmark datasets in the future.

## Acknowledgments and Disclosure of Funding

This work is supported by the Hong Kong RGC Projects (No. 12200424), the RMGS project (Artificial Intelligence and Big Data Analytics for Social Good), and the National Natural Science Foundation of China (No. 62472304 and No. 62436001). The opinions expressed in this paper are those of the authors and do not necessarily reflect the views of the funding agencies. We would like to express our gratitude to the anonymous reviewers for their valuable feedback.

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

Figure 11: Hyperparameter analysis of information diffusion predictor on two real-world datasets.

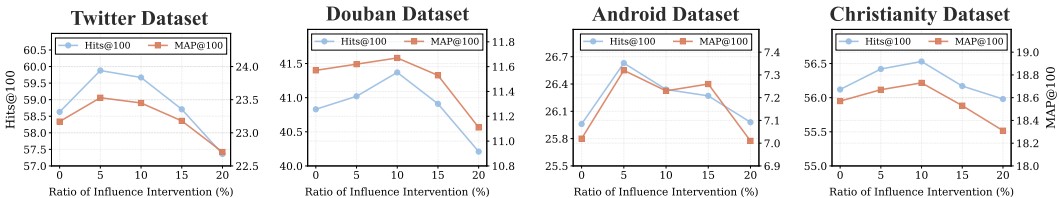

Figure 12: Influence intervention on four widely-used basic datasets.

# A   Additional Evaluation

## A.1   Sensitivity Analysis

As illustrated in Figure 11, we conduct a comprehensive sensitivity analysis on two key hyperparameters using both the *News* and *Weibo* datasets, which are *Temperature Factor* $\alpha$ and *(2) Number of GNN Layers L*. Each hyperparameter is varied independently while the others remain fixed. Performance trends are evaluated to inform robust settings. *(1) Temperature Factor $\alpha$:* This parameter, introduced in Eq. 10, modulates the sharpness of attention weights based on the influence graph. A larger $\alpha$ emphasizes the strongest influence connections by increasing the softmax contrast, potentially suppressing secondary or implicit correlations—such as shared preferences among users in the same cascade—that may still contribute valuable contextual signals. While emphasizing dominant influence paths is beneficial to a certain extent, excessive $\alpha$ values risk discarding useful structural information and thereby degrading performance. On the other hand, very small $\alpha$ leads to overly uniform attention, reducing the model's ability to distinguish key influencers. Empirical results indicate that setting $\alpha = 1$ provides a balanced calibration, effectively preserving both focused attention and peripheral contextual relevance. *(2) Number of GNN Layers L:* We employ a classical GAT encoder for social representation learning, and examine how the number of GNN layers $L$ affects performance. Deeper GNNs aggregate information from more distant neighbors, which can, in principle, capture higher-order structural signals and extend homophily-based reasoning. However, increasing $L$ introduces the risk of over-smoothing, where node embeddings become indistinguishable and less informative. In our experiments, a single GAT layer ($L = 1$) consistently yields the best results. This suggests that in diffusion-centered networks, the most relevant information is often localized within the closest neighborhood, which represents frequent co-participants in cascades, thus making deeper aggregation unnecessary and potentially harmful. We note that other hyperparameters (e.g., hidden dimensionality, learning rate) follow expected trends and exhibit limited sensitivity. Their configurations are reported in the implementation details.

## A.2   Empirical Analysis on Influence

In this paper, we use LLMs to analyze the key influences in information diffusion. Widely-used benchmark datasets do not include any textual data for advanced LLMs, and only contain a social network structure and basic cascade metadata (participant IDs and timestamps). Therefore, we analyze these datasets here, as in Section 4.3, to verify our observation that there is an imbalanced diffusion ability for different social actors within cascades. To empirically assess the necessity and robustness of the causal influences, we introduce a controlled intervention experiment. Specifically, for a given cascade, we randomly select a fraction of participant-to-participant influence edges and set their corresponding attention scores to $-\infty$, eliminating these links from contributing to the diffusion dynamics modeled by the attention mechanism. We then evaluate how this partial influence masking affects the performance of the downstream diffusion prediction task. As illustrated in Figure 12, we observe a counter-intuitive but insightful result: **moderate levels of influence masking, particularly at intervention ratios below 15%, do not degrade model performance.** In fact, across all four benchmark datasets, **slight performance improvements** are consistently observed under such interventions. This phenomenon suggests that many influence relations in social cascades are either spurious or redundant. While some users play critical roles

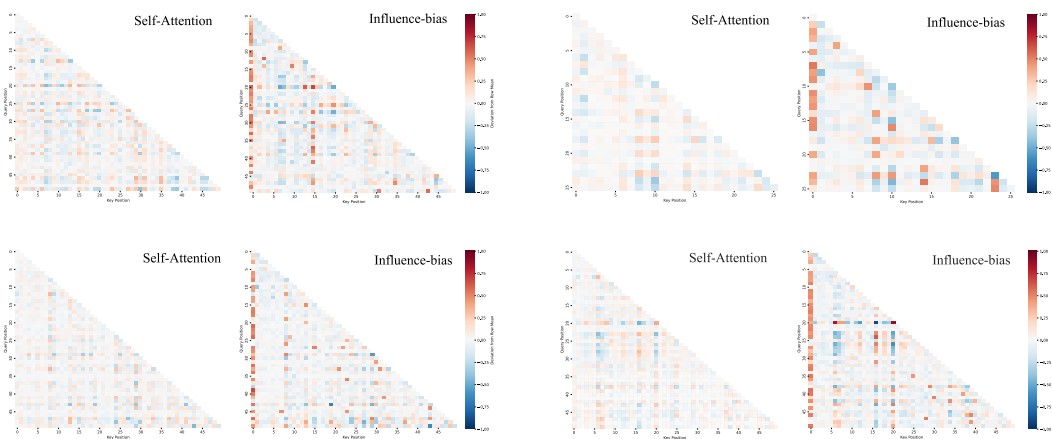

Figure 13: Attention Visualization (Q_K): Self-Attention *vs.* our MILD with influence bias.

Table 4: Performance on information diffusion prediction. The higher scores are better.

| Datasets Metrics | News | | | | | | Weibo | | | | | |
|---|---|---|---|---|---|---|---|---|---|---|---|---|
| | H@10 | H@50 | H@100 | M@10 | M@50 | M@100 | H@10 | H@50 | H@100 | M@10 | M@50 | M@100 |
| DyHGCN [37] | 18.94 | 24.50 | 27.31 | 10.50 | 10.76 | 10.80 | 10.51 | 15.39 | 18.50 | 6.01 | 6.23 | 6.27 |
| Standard Deviation | ± 1.12 | ± 1.89 | ± 1.45 | ± 0.33 | ± 0.41 | ± 0.36 | ± 0.56 | ± 0.66 | ± 0.74 | ± 0.42 | ± 0.47 | ± 0.35 |
| MSHGAT [28] | 20.10 | 25.82 | 28.85 | 11.68 | 11.95 | 11.99 | 11.41 | 18.34 | 21.77 | 6.16 | 6.48 | 6.53 |
| Standard Deviation | ± 0.98 | ± 1.25 | ± 1.03 | ± 0.47 | ± 0.43 | ± 0.39 | ± 0.72 | ± 0.67 | ± 0.75 | ± 0.34 | ± 0.42 | ± 0.39 |
| DisenIDP [3] | 20.47 | 26.33 | 29.08 | 12.03 | 12.29 | 12.36 | 12.04 | 18.83 | 22.61 | 6.73 | 7.01 | 7.06 |
| Standard Deviation | ± 0.50 | ± 0.76 | ± 0.88 | ± 0.13 | ± 0.09 | ± 0.14 | ± 1.37 | ± 1.24 | ± 1.18 | ± 0.62 | ± 0.47 | ± 0.55 |
| RotDiff [23] | 20.95 | 27.02 | 29.80 | 12.52 | 12.80 | 12.84 | 12.99 | 20.46 | 24.70 | 7.76 | 8.12 | 8.18 |
| Standard Deviation | ± 1.30 | ± 1.12 | ± 0.98 | ± 0.55 | ± 0.69 | ± 0.63 | ± 0.81 | ± 1.14 | ± 1.20 | ± 0.68 | ± 0.79 | ± 0.82 |
| MINDS [14] | 20.04 | 25.66 | 28.70 | 11.72 | 11.93 | 11.98 | 11.69 | 18.07 | 21.66 | 6.24 | 6.51 | 6.57 |
| Standard Deviation | ± 0.89 | ± 1.10 | ± 1.06 | ± 0.25 | ± 0.30 | ± 0.28 | ± 1.25 | ± 1.54 | ± 1.67 | ± 0.61 | ± 0.78 | ± 0.71 |
| MGCL [5] | 21.26 | 28.11 | 31.93 | 12.90 | 13.18 | 13.24 | 13.04 | 20.18 | 24.75 | 7.60 | 7.99 | 8.03 |
| Standard Deviation | ± 1.67 | ± 1.58 | ± 1.72 | ± 0.54 | ± 0.48 | ± 0.63 | ± 1.40 | ± 1.92 | ± 1.88 | ± 0.31 | ± 0.37 | ± 0.35 |
| GODEN [29] | 22.01 | _29.14_ | _33.17_ | 13.34 | 13.67 | 13.73 | _14.08_ | 21.95 | _26.71_ | _8.09_ | _8.28_ | _8.52_ |
| Standard Deviation | ± 1.15 | ± 1.29 | ± 1.68 | ± 0.86 | ± 0.79 | ± 0.90 | ± 1.34 | ± 1.53 | ± 1.44 | ± 0.67 | ± 0.66 | ± 0.61 |
| CARE [41] | _22.47_ | 28.71 | 32.01 | _14.53_ | _14.81_ | _14.85_ | 13.35 | _21.96_ | 26.62 | 7.65 | 8.05 | 8.11 |
| Standard Deviation | ± 1.16 | ± 1.38 | ± 1.33 | ± 0.68 | ± 0.74 | ± 0.76 | ± 1.58 | ± 1.82 | ± 1.79 | ± 0.37 | ± 0.34 | ± 0.35 |
| **MILD (Ours)** | **24.07** | **31.33** | **35.52** | **15.50** | **15.85** | **15.91** | **14.71** | **23.37** | **28.17** | **8.71** | **9.10** | **9.17** |
| Standard Deviation | ± 1.10 | ± 0.99 | ± 1.04 | ± 0.84 | ± 0.61 | ± 0.67 | ± 1.23 | ± 1.47 | ± 1.26 | ± 0.69 | ± 0.53 | ± 0.52 |

in propagating information, others are likely peripheral participants whose modeled connections may introduce noise rather than signal. These findings strongly support our core hypothesis: **the effectiveness of modeling information diffusion can be substantially improved by identifying and prioritizing key influences while suppressing inconsequential ones.** This experiment serves as an empirical validation of our influence-centric design: a fine-grained representation of selective influence, rather than a dense or fully-connected modeling assumption, yields a more faithful and efficient representation of real-world information diffusion dynamics.

### A.3 Experiment statistical significance

The dataset is divided according to the chronological order of events and cannot be shuffled. In real applications, we can only use past information diffusion events to predict future events and avoid label leakage. Therefore, we randomly selected five seeds for statistical significance. We clearly calculate the standard deviation. As shown in Table 4, the experimental results show that our MILD consistently and stably outperforms eight baselines.

## B Algorithm Details

### B.1 Potential diffusion path search

The detailed diffusion path search algorithm can be found in Algorithm 1. Lines 1-2 conduct the single-source BFS algorithm with the original participant $v_1$ as the single source, which returns the BFS tree $T_{\text{BFS}}$. $T_{\text{BFS}}$ is composed of shortest paths between $v_1$ and $v \in V_C$, satisfying (D1) and (D2). For each shortest path in $T_{\text{BFS}}$, the algorithm filter the long shortest path $\psi$ with length greater than $\ell$ (Lines 3-4). Considering the efficiency,

---

**Algorithm 1** Potential Diffusion Path Search.

---

**Require:** A cascade graph $G_C = (V_C, E_C)$, the graph density constraint $r$, the threshold of shortest path $\ell$.

**Ensure:** $\psi \in \Psi$ satisfies the requirements (D1), (D2) and (D3).

1: Select the original participant $v_1$ as the source node;
2: $T_{\text{BFS}} \leftarrow$ shortest paths between $v_1$ and $v_i \in V_C \setminus v_1$ by single-source BFS;
3: **for** $\psi \in T_{\text{BFS}}$ **do**
4:    **if** $|\psi| \geq \ell$ **then**
5:       Calculate the graph density of $G_C$ by $\hat{d} = |V_C|/|E_C|$; (Details in Appendix B.3)
6:       Select the source $s$ and the target $\tau$ in $\psi$;
7:       **if** $\hat{d} \geq r$ **then**
8:          $\{\psi\}_{simple} \leftarrow \{\psi'|v_i \neq v_j, \forall v_i, v_j \in \psi', \psi' \in G_C\}$ from $s$ to $\tau$ by DFS;
9:          $\Psi \cup \{\psi\}_{simple}$;
10:      **else**
11:         $\{\psi\}_{shortest} \leftarrow$ select all shortest paths from $s$ to $\tau$ by BFS;
12:         $\Psi \cup \{\psi\}_{shortest}$;
13:    **else**
14:       **if** $\forall v_i \in \psi$ and $v_i \notin \Psi$ **then**
15:          $\Psi \cup \psi$;
16: **return** $\Psi$;

---

Line 5 calculates the graph density $\hat{d}$ of $G_C$ based on analysis in Appendix B.3 to control the time complexity of searching all possible paths. Firstly, the algorithm selects the source $s$ and target $\tau$ of the filtered $\psi$. Then, conduct the DFS algorithm to find all simple paths between $s$ and $\tau$ if $\hat{d} \geq r$ and add them into $\Psi$ (Lines 7-9). If $\hat{d} \leq r$, conduct the BFS algorithm to find all shortest paths between $s$ and $\tau$ if $\hat{d} \geq r$ and add them into $\Psi$ (Lines 10-12). If $\psi$ is a short shortest path, add $\psi$ into $\Psi$ if any nodes in $\psi$ are excluded by $\Psi$ (Lines 13-15). Finally, line 16 returns the satisfied $\Psi$.

## B.2   User Influence and Activity Levels Estimation

The estimation process of user influence and activity levels can be divided into four steps: **key user attributes selection** from user profiles, **min-max normalization** of the selected attributes, **influence and activity values estimation**, and **assigning specific levels** based on the estimated values.

**1st step: key user attributes selection from user profiles.** Based on the well-known communication theory, we select the key user attributes. The details are as follows.

- **Follower Number ($x_1$)** directly reflects a user's network centrality, aligning with the scale-free feature described in complex network theory (Barabási-Albert model)[2]. Users with a high number of followers act as hub nodes that can trigger cascaded propagation, and the influence range follows a power-law distribution.
- **Bi-followers Number ($x_2$).** Based on the theory of strong ties (Granovetter) [7], mutual relationships represent bidirectional trust links, and such connections have a higher conversion rate in information diffusion.
- **Published Statuses Number ($x_3$)**, which is in accordance with the attention economy model [4]. High-frequency publishers occupy users' attention bandwidth through content streams, thereby establishing a temporal advantage.
- **Verified as an influential user by the online platform ($x_4$).**

**2nd step: Min-Max Normalization of the selected attributes.** For each of the above selected attributes, we normalize the value across all users based on min-max normalization as follows.

$$x_{scaled} = \frac{x - x_{min}}{x_{max} - x_{min}} \tag{11}$$

**3rd step: Influence and activity estimation.** Based on the normalized attributes, we calculate the influence and activity values of each user. For influence values, we assign the normalized value $x_1$ of the follower number as the influence value. For activity value, we calculate it as follows.

$$f(x) = w_2 x_2 + w_3 x_3 + w_4 x_4, \tag{12}$$

where $w_2, w_3, w_4$ are the hyperparameters designed based on the above communication theories, which are $w_2 = 0.1, w_3 = 0.8, w_4 = 0.1$, respectively.

**4th step: Assigning specific levels based on the estimated values.** We assign specific levels for each estimation, i.e., 0-20%: poor, 20%-40%: low, 40%-80%: moderate, 80%-95%: high, 95%-100%: outstanding.

### B.3 Graph Density Calculation in Diffusion Path Search Algorithm

The graph density calculation in Algorithm 1 follows E. Lawler (1976), *Combinatorial Optimization: Networks and Matroids* [17]. The formulation is:

$$\text{graph density} = \frac{|E|}{|V|}, \tag{13}$$

where $|E|$ is the number of edges) and $|V|$ is the number of nodes. This calculation formulation is closely relevant to the time complexity analysis in Section 3.2.

## C  Theoretical Analysis

### C.1  Back-door Adjustment of Existing Methods

According to the SCM, the statistical distribution of $P(Y|C)$ has following mathematical expression,

$$P(Y|C) = \sum_{u \in U} P(u|C)P(Y|C,u) = \mathbb{E}_{P(u|C)}P(Y|C,u), \tag{14}$$

The key is to capture the true causal relationship between $C$ and $Y$. From the perspective of causal inference, we can achieve this via modeling $P(Y|do(C))$, where $do(C)$ denotes the do-calculus operation and introduces an intervention on $C$. Specifically, $do(C)$ removes links from $U$ to $C$ and blocks associative paths from $C$ to $Y$ apart from the direct causal one $C \to Y$. We employ the basic back-door adjustment to estimate $P(Y|do(C))$ as below.

$$P(Y|do(C)) = \sum_{u \in U} P(u)P(Y|C,u). \tag{15}$$

However, the above equation does not satisfy the backdoor condition since the associations between both $C \leftarrow U$ and $U \to Y$ are unobserved. Thus, the variable $U$ is unobserved and implicit. Hence, $U$ cannot be used to block the backdoor path from $C$ to $Y$. There is no way to estimate $P(u)$ and $P(Y|C,u)$. The causal effect of the historical cascade on future participants is not identifiable in this model.

### C.2  Front-Door Adjustment of Our Method

We add the transparent diffusion influence as the intermediate variable $Z$ between $C$ and $Y$ as shown in Figure 2. It satisfies the following front-door criterion: (C1) $Z$ intercepts all directed paths from $C$ to $Y$, (C2) there is no unblocked back-door path from $C$ to $Z$, and (C3) all backdoor paths from $Z$ to $Y$ are blocked by $C$. The (C1) condition requires $Z$ to represent the nature of the cascade that provides necessary and sufficient information for predicting the next participants $Y$, and the (C2) and (C3) conditions suggest that only the observed cascade sequence $C$ can be used to estimate the causal relationship $C \to Z$, as the backdoor path from $Z$ to $Y$, namely $Z \leftarrow C \leftarrow U \to Y$, can be blocked by conditioning on $C$.

Based on the above conditions, we can formulate our model by chaining together the two partial effects and summing over all states $z$ of $Z$ as follows If the do-calculus is performed on $Z$, the probability of $Y$ is $P(Y|do(Z))$. Then, if the do-calculus is performed on $X$, the probability of conducting the do-calculus of $Z$ is $P(Z|do(x))$.

$$P(Y|do(C)) = \sum_{z \in Z} P(Y|do(Z))P(Z|do(C)) \quad (C1) = \sum_{z \in Z} P(Y|do(Z))P(Z|C) \quad (C2)$$

$$= \sum_{z \in Z} \sum_{c \in C'} P(Y|Z,C')P(C')P(Z|C) \quad (C3), \tag{16}$$

where $C'$ contains all observed cascade sequences in training data, $z$ denotes the specific diffusion influence matrix in $Z$.

### C.3  MILD Under Causal Framework

MILD is strictly designed to satisfy three front-door criteria in the causal framework to make a more accurate diffusion prediction. (C1) $Z$ intercepts all directed paths from $C$ to $Y$. When $Z$ can represent the information

diffusion itself, $Z$ can intercept all directed paths from $C$ to $Y$. To achieve it, MILD captures almost all necessary and sufficient diffusion information to derive $Z$ based on communication theories to make sure that $Z$ can represent the nature of the information cascade. (C2) there is no unblocked back-door path from $C$ to $Z$, and (C3) all backdoor paths from $Z$ to $Y$ are blocked by $C$. MILD extract observed information cascades from the real-world social platform as $C$ to block the backdoor path from $Z$ to $Y$, which is $Z \leftarrow C \leftarrow U \rightarrow Y$, satisfying (C2) and (C3).

## C.4  Do-operation analysis of MILD

$P(Y|do(C))$ contains two do-operations, which can be formulated as $P(Y|do(C)) = \sum_{z \in Z} P(Y|do(Z))P(Z|do(C))$ based on the front-door criterion (C1). In MILD, $do(C)$ is to select several users who can form an observed information cascade, forcing $C = c$. $do(Z)$ operation is to derive diffusion influence representation $z$ based on given interventions, which are designed potential diffusion paths $\psi$, temporal order $T$, user influence and activity levels $O$, and comments $B$ from specific cascade $c$, forcing LLM to derive $Z = z$. In the $Z$ space, given the observed cascade $c$ without interventions, $Z$ contains all possible random diffusion processes. Thus, $P(Z|do(C))$ can be assumed as $P(z|do(C)) = \frac{1}{|Z|}, z \in Z$. With interventions of $Z$, $P(Y|do(Z))$ is the probability of Transformer prediction $Y$ conditioned on our derived diffusion influence graph $z = G_I$.

## C.5  NP-Complete Proof of Hamiltonian Path Problem

**Problem Definition.** A Hamiltonian Path in a graph is a path that visits each vertex exactly once. Given a graph $G = (V, E)$, the Hamiltonian Path Problem (HPP) asks whether there exists such a path in $G$. If the path starts and ends at the same vertex, it is called a Hamiltonian Cycle.

**Proof.** As with any NP-completeness proof, this one has two parts. First, we will go over why Hamiltonian Path is in the NP class. To be in NP means that, given a proposed solution (a certificate) to the problem, we can verify the solution in polynomial time. In this case, that means that given a directed graph $G$ and path between 2 vertices, we can check in polynomial time if the path is a Hamiltonian Path.

**Step 1: Proving Hamiltonian Path is in NP.** To show that HPP is in NP, we need to prove that a given solution can be verified in polynomial time. Suppose we are given a certificate (a sequence of vertices). Then, we need to check two constraints: (1) The sequence contains all vertices exactly once. (2) There exists an edge between each consecutive vertex in the sequence. For time complexity analysis, if we check edges in an adjacency matrix, the verification takes at most $O(|V|^2)$ time. While, if we check edges in adjacency lists, the verification takes at most $O(|V| + |E|)$. Thus, we can know that the verification can be processed in polynomial time. Since the verification is polynomial, HPP belongs to NP.

**Step 2: Proving NP-Hardness.** To prove HPP is NP-Hard, we reduce a known NP-Complete problem to it. A standard reduction is from the Hamiltonian Cycle Problem (HCP), which is already NP-complete. Thus, we give the following explanations of the reduction from Hamiltonian Cycle to Hamiltonian Path. Given a graph $G = (V, E)$, we need to check it for a Hamiltonian Cycle. The goal of the reduction is the transformation of modifying $G$ to construct a new graph $G'$. First, we pick an arbitrary vertex $v \in V$. Then, we remove $v$ from $G$, resulting in graph $G'$. After transforming $G$, we can claim as follows,

*Claim:* $G$ has a Hamiltonian Cycle if and only if $G'$ has a Hamiltonian Path.

Here, we prove this claim and analyze that this reduction can be done in polynomial time. If $G$ has a Hamiltonian Cycle, removing $v$ will break this Hamiltonian Cycle into a Hamiltonian Path. Conversely, if $G'$ has a Hamiltonian Path, adding $v$ back can reconstruct a Hamiltonian Cycle. This transformation is done in polynomial time, proving that HPP is at least as hard as HCP. Since HCP is NP-Complete, and we have reduced it to HPP in polynomial time, it follows that HPP is NP-Hard.

We prove HPP is in the NP class and it is NP-hard. Thus, HPP is an NP-complete problem.

## C.6  NP-hard Proof of All Simple Path Problem

**Problem Definition.** Given a graph $G = (V, E)$, a simple path is a path in $G$ that does not repeat any vertex. Given two random vertices in $G$ as the source $s$ and target $\tau$, find all possible simple paths between $s$ and $\tau$ in $G$.

**Proof.** To prove that All Simple Path Problem (ASPP) is an NP-hard problem, we reduce a proved NP-complete problem to it. As proved in Appendix C.5, the Hamiltonian Path is an NP-complete problem. Thus, we give the following explanations of the reduction from the Hamiltonian Path to All Simple Paths. Given a graph $G = (V, E)$, we need to check it for a Hamiltonian Path.

*Claim:* If you could find all simple paths between two nodes, you could determine if there is a Hamiltonian path between them by checking if any of the paths visit all the vertices.

We explain this claim as follows. First, we construct an arbitrary instance of HPP. We choose any two vertices $s$ and $t$ in $G$. Then, we determine if there is a Hamiltonian path starting at $s$ and ending at $t$. Second, we list all simple paths from $s$ to $t$ and check if any of these paths visit all vertices exactly once. If such a path exists, it is a Hamiltonian path. The ability to find all simple paths allows us to solve the Hamiltonian Path problem by simply checking the length of each path. Hence, solving the problem of finding all simple paths would solve the Hamiltonian Path problem. This reduction from HPP can be done in polynomial time, which proves that ASPP is NP-hard, as HPP is a proven NP-complete problem.

### C.7 NP-hard Proof of Diffusion Path Search Problem

**Problem definition.** Given a cascade graph $G_C = (V_C, E_C)$ with one unique root $v_1$, the problem returns a collection of path structures $\Psi \subseteq G_C$. There are three constraints. (D1): all paths $\psi \in \Psi$ start from the unique root $v_1$. (D2): $V_\Psi$ covers all nodes in $V_C$. (D3): $\Psi$ contains all possible path between $v_1$ and $v \in \{v_i | dist(v_1, v_i) \geq \ell\}$.

As all paths in this paper are non-loop paths with no repeated nodes, the Diffusion Path Search Problem (DPSP) is similar to the All Simple Paths Problem (ASPP). The proof of NP-hardness of DPSP is also similar to the proof of ASPP, which is in Appendix C.6.

**Proof.** To prove that DPSP is an NP-hard problem, we reduce the proved NP-complete problem to it. As proved in Appendix C.5, the Hamiltonian Path is an NP-complete problem. If we can prove that a polynomial-time reduction exists from the Hamiltonian Path Problem (HPP) to the Diffusion Path Search Problem (DPSP), we prove the DPSP is an NP-hard problem. Thus, we give the following explanations of the reduction from the Hamiltonian Path to the Diffusion Path. Given a graph $G = (V, E)$, we need to check it for a Hamiltonian Path.

*Claim:* If you could find a diffusion path set $\Psi$ between two nodes $s$ and $\tau$, you could determine if there is a Hamiltonian path between them by checking if any of the path in $\Psi$ visit all the vertices.

We explain this claim as follows. First, we construct an arbitrary instance of HPP. We choose the same graph $G_C$ and two vertices $s$ and $t$ in $G_C$. Second, we use an algorithm for the diffusion path search problem to list all potential diffusion paths in $\Psi$ between $s$ and $t$. Then, we check if any enumerated path in $\Psi$ has length $n - 1$ (where $n = |V_C|$). If such a path exists, $G_C$ contains a Hamiltonian path. Otherwise, it does not.

Based on the above steps, we can find that the reduction requires no modification to $G$, so it runs in $O(1)$ time. The reduction can be done by simply checking the length of each path. Thus, it is a polynomial-time reduction. While enumerating all paths may take exponential time, the reduction assumes the existence of a polynomial-time diffusion influence path search problem oracle. If such an oracle existed, solving the Hamiltonian Path Problem would also take polynomial time (contradicting its NP-completeness). Thus, the diffusion influence path search problem must be NP-hard.

### C.8 Diffusion Influence Degree Proof in Robustness Evaluation

Here, we give the proof of the path diffusion influence degree.

**Proof.** Based on the chain rule of conditional probability, given a diffusion path $\psi = (v_1, v_2, ..., v_n)$ with $n$ nodes, we can easily obtain the diffusion influence degree of the whole path $\psi$, which can be formulated as

$$P(\psi) = \prod_{1 < i \leq n} P(v_i | v_{i-1}, ..., v_1) P(v_1). \tag{17}$$

An information diffusion path indicates the predecessor passing the information to the successor. For example, given a diffusion path $\langle v_1.v_2, v_3 \rangle$, $v_1$ passes the diffusion influence to $v_3$ solely through $v_2$ within this path, which means $v_1$ and $v_3$ are conditionally independent on $v_3$. Generally, $v_{i-1}$ and $v_{i+1}$ are conditionally independent on $v_i$. Thus, the Equation 17 can be simplified as

$$P(\psi) = \prod_{1 < i \leq n} P(v_i | v_{i-1}, ..., v_1) P(v_1) = \prod_{1 < i \leq n} P(v_i | v_{i-1}) P(v_1). \tag{18}$$

$\square$

## D  Details of Experimental Setup

### D.1  Datasets

We conduct our experiments on two real-world datasets harvested from Weibo [38, 1], a leading Chinese microblogging platform. We refer to these two corpora as **News** and **Weibo**, defined as follows:

- **News.** This dataset consists of all repost cascades initiated by the official Weibo account "Daily News", social network structures and user profiles. Each cascade captures the reposting timeline, user interactions, and comment threads triggered by a single news post.
- **Weibo.** This dataset contains social network structures, user profiles, and numerous repost cascades published by five randomly chosen opinion leaders on the Weibo platform, one from each domain: sports, entertainment, technology, society, and lifestyle. These hub influencers exhibit heterogeneous diffusion patterns due to their distinct audience profiles.

Here, we detail the data preprocessing methods applied to News and Weibo, focusing primarily on user profile features, social graph construction, and cascade formation.

**User Profile Features**   For every user involved in a cascade, we extract the following 13 profile attributes:

| | |
|---|---|
| `bi_followers_count` | Number of mutual followers |
| `verified_status` | Boolean, official verification flag |
| `followers_count` | Total follower count |
| `location`, `province`, `city` | Self-declared geographic metadata |
| `friends_count` | Number of followees |
| `name`, `gender` | Display name and gender |
| `created_time` | Account creation timestamp |
| `verified_type` | Verification category (e.g. personal, media) |
| `statuses_count` | Total number of posts published |
| `description` | Self-written bio text |

**Social Graph Construction**   We model the "following" relations among users as a directed graph $\mathcal{G} = (\mathcal{V}, \mathcal{E})$, where each node $v \in \mathcal{V}$ is a user and each edge $(u \to v) \in \mathcal{E}$ indicates that $u$ follows $v$. This graph underpins all diffusion processes we analyze.

**Cascade Formation**   Each original post is treated as an *information unit*. To assemble a diffusion cascade, we record for every participant: (1) The type of behavior (repost, comment, like); (2) The timestamp of the behavior; (3) Any textual content of comments. These elements yield a temporally ordered cascade $\mathcal{C} = \{(u_i, t_i, a_i)\}_{i=0}^{|\mathcal{C}|-1}$, where $u_i$ is the user, $t_i$ the action time, and $a_i$ the action type.

Table 5: Statistics of the four widely-used datasets.

| | Twitter | Douban | Android | Christianity |
|---|---|---|---|---|
| # Users | 12,627 | 12,232 | 9,958 | 2,897 |
| # Links | 309,631 | 396,580 | 48,573 | 35,624 |
| Density (%) | 24.52 | 30.21 | 4.87 | 12.30 |
| # Cascades | 3,442 | 3,475 | 679 | 589 |
| Avg. Len. | 32.60 | 21.76 | 33.30 | 22.90 |
| Density (%) | 8.89 | 6.18 | 2.27 | 4.66 |

In addition to our News and Weibo datasets, in the supplemented empirical evaluation, we benchmark against four widely used public diffusion collections, including Twitter [9], Douban [40], Android [24], and Christianity [24]. Each dataset comprises an underlying social network and a set of basic cascades, where each cascade records only user IDs and their timestamps. Table 5 presents the descriptive statistics for these benchmarks.

## D.2   Competitors

Most existing methods for diffusion prediction adopt a two-phase framework: *GNNs + sequence models*. In this setting, graph neural networks learn structural representations from social or diffusion graphs, and sequential models (e.g., RNNs or transformers) model cascade dynamics over time. We summarize the representative baselines as follows:

- **DyHGCN** [37] builds a dynamic heterogeneous graph by combining the social network with historical diffusion paths. A heterogeneous GCN captures time-sensitive node embeddings, followed by multi-head self-attention to model cascade-level dependencies.
- **MS-HGAT** [28] represents diffusion history using timestamped hypergraphs and learns user embeddings through hierarchical attention. It integrates static social links with dynamic cascades using gated memory modules.
- **DisenIDP** [3] disentangles user intent by constructing dual hypergraph networks. Intent-specific embeddings are generated and aligned using self-supervised losses, allowing fine-grained modeling of diffusion factors.

- **RotDiff** [23] maps users into a hyperbolic space, where Lorentzian rotations encode asymmetric influence patterns. Rotated self-attention captures hierarchical dependencies for cascade prediction.
- **MINDS** [14] introduces multi-scale hypergraph learning to jointly capture micro- and macro-level cascade patterns. An adversarial alignment module ensures cross-scale consistency.
- **MGCL** [5] applies contrastive learning to suppress noise from low-quality graphs. Multiple structural views are generated, and agreement is enforced via contrastive objectives to enhance embedding quality.
- **GODEN** [29] formulates diffusion as a continuous-time process using neural ODEs. Two coupled ODEs evolve node and edge states, which are fused with timestamp-aware attention for temporal prediction.
- **CARE** [41] enhances prediction by retrieving semantically similar historical cascades. These retrieved instances are encoded via a transformer to support retrieval-augmented diffusion modeling.

## D.3 Evaluation Metrics

To assess the accuracy of predicting the next participant in a diffusion cascade, we adopt two widely-used ranking metrics of the diffusion prediction task: *Hits@K* and *Mean Average Precision@K* (MAP@K). Hits@K measures the recall at a fixed cutoff, i.e., the fraction of cascades for which the true participant appears among the top $K$ predictions. While MAP@K further penalizes late placements by weighting each hit by the inverse of its rank, thus evaluating the quality of the ordering within the top-$K$ list.

Formally, let $N$ be the number of test cascades, and for each cascade $i$ let $u_i$ denote the ground-truth next participant and

$$\hat{U}_i = \left[\hat{u}_{i,1}, \hat{u}_{i,2}, \ldots, \hat{u}_{i,K}\right]$$

be the ordered list of top-$K$ predictions. Define the rank function

$$\text{rank}(u_i) = \begin{cases} r, & \text{if } u_i = \hat{u}_{i,r} \text{ for some } r \leq K, \\ \infty, & \text{otherwise,} \end{cases}$$

and the indicator $\mathbf{1}[\cdot]$. Then:

$$\text{Hits@}K = \frac{1}{N} \sum_{i=1}^{N} \mathbf{1}\left[\text{rank}(u_i) \leq K\right], \tag{19}$$

$$\text{MAP@}K = \frac{1}{N} \sum_{i=1}^{N} \frac{\mathbf{1}\left[\text{rank}(u_i) \leq K\right]}{\text{rank}(u_i)}. \tag{20}$$

By combining these two metrics, we capture both the ability to include the true user within a limited prediction budget and the precision with which it is ranked among the top candidates.

# E Prompt Design

**Prompt design in MILD.** We design the prompt to instruct the LLM to analyze the diffusion influence between users within each cascade. The potential diffusion path $\psi \in \Psi$, temporal order $T$, descriptions of user influence and activity levels $O$, and comments $B$ are integrated into the designed template $\Phi(\cdot)$, forming prompts $\Phi(\psi, T, O, B)$ to provide sufficient information for analysis. To guarantee the reliability of the derivation, we add common sense of information diffusion to the design of the instruction. For instance, the latter participants do not influence the earlier participants. The detailed prompt can be found in Appendix Figure 14.

**Prompt design in measurements of $G_I$.** In Robustness Evaluation, we construct real and fake diffusion paths within cascades, and design the misleading instructions to lead the LLM into believing that all paths are real to analyze the influence degree. For each cascade, we create multiple real and fake diffusion paths. We randomly select one user out of the real path to replace one user in the real path to create the fake diffusion path and iterate it several times. We then mix all paths as real diffusion paths. Then we instruct the LLM to analyze the diffusion degrees of each edge in each path. The detailed prompt can be found in Appendix Figure 15.

**Prompt design in ablation studies.** We conduct ablation studies to evaluate the necessity of the four key diffusion influencing factors. For each ablation study, we remove the relevant factor from the prompt. For instance, if we evaluate the necessity of temporal order, we change the original prompt $\Phi(\psi, T, O, B)$ to $\Phi(\psi, O, B)$ as the prompt for the ablation study of the temporal order. The detailed prompt can be found in Appendix Figure 16. The prompts without user profiles and shared comments are similar to the prompt without temporal order. Particularly, for the ablation study without our potential diffusion pathways, we use all shortest paths in $T_{\text{BFS}}$ to replace the potential diffusion paths in the original as $\Phi(\psi*, T, O, B)$, where $\psi*$ denotes the shortest path. The detailed algorithm can be found in Section 3.2 and Algorithm 1.

## Figure 14: Example Prompt for Diffusion Influence Deriving

**SYSTEM**
**[English Version]** Please identify all possible pairs of users that may have influence, considering the chronological order of replies, potential influence pathways, user influence and activity, and user reply content. Note: 1. Users who reply earlier in the chronological order will not be influenced by later respondents; 2. Only the influence between the given user IDs will be analyzed. Use a standard list format, and do not provide any text. Respond with pairs of user IDs in the following format: [[..,..],..].
**[Chinese Version]** 请你找出所有可能存在影响的用户对，结合回复时间顺序，潜在影响路径，用户影响力和活跃度，用户回复内容。注意1.在回复时间顺序中更早回复的用户不会被更晚的用户影响，2.只分析给定user id之间的影响。使用标准list格式，禁止回答任何文字，用user id组成用户对，回答格式如下：[[..,..],..]。

**USER**
**[English Version]**
**Reply timeline based on user IDs:** '['4659', '28931', '6603']'.
**Potential influence path:** '['4659', '6603', '28931']'.

**User influence and activity levels:**
User 4659: outstanding influence, outstanding activity level.
User 6603: high influence, high activity level.
User 28931: high influence, moderate activity level.

**Comments published by each user:**
4659: "Yi Siling wins the first gold medal at the London Olympics."
6603: "Watching this made my heart race! [Touched by the victory] [Thumbs up] [Thumbs up]."
28931: "Especially that moment of overtaking."

**[Chinese Version]**
由用户**id**组成的回复时间顺序**:**['4659', '28931', '6603']。
潜在影响路径**:**['4659', '6603', '28931']。

用户影响力和活跃度:
4659:影响力高,活跃度高。6603:影响力较高,活跃度较高。28931:影响力较高,活跃度中等。

每个用户发表内容**:**
4659说伦敦奥运会易思玲夺首金。
6603说看的我心跳超快[夺冠感动][赞][赞]。
28931说尤其反超那一刻。

**ASSISTANT**
[[4659, 6603], [4659, 28931], [28931, 6603]]

---

Figure 15: Example Prompt without Temporal Order

**SYSTEM**
Please identify all possible pairs of users that may have influence, considering potential influence pathways, user influence and activity, and user reply content. Note: Only the influence between the given user IDs will be analyzed. Use a standard list format, and do not provide any text. Respond with pairs of user IDs in the following format: [[..,..],..].

**USER**
**Potential influence path:** '['4659', '6603', '28931']'.

**User influence and activity levels:**
User 4659: outstanding influence, outstanding activity level.
User 6603: high influence, high activity level.
User 28931: high influence, moderate activity level.

**Comments published by each user:**
4659: "Yi Siling wins the first gold medal at the London Olympics."
6603: "Watching this made my heart race! [Touched by the victory] [Thumbs up] [Thumbs up]."
28931: "Especially that moment of overtaking."

**ASSISTANT**
$[[4659, 6603], [6603, 28931]]$

---

Figure 16: Example Prompt for True and Fake Diffusion Paths

**SYSTEM**
Please analyze the diffusion influence degree for each edge in the diffusion path, with a value range of [0,1]. The difference in propagation influence degrees within the same propagation path should not exceed 0.2. Carefully examine whether the content published by intermediate users discusses the same topic, as there may be false propagation paths. Respond in standard JSON format, without any additional text, as follows: [{edge : [..., ...], influence_value : ...}, ...]

**USER**
**Potential influence path:** '['4659', '17264', '11799']'.

**Comments published by each user:**
4659: "Lin Dan staged a dramatic comeback against Lee Chong Wei to claim China's 26th gold medal."
17264: "Woooahhh[tears] Such an emotional victory! This was so hard-fought!"
11799: "[Thumbs up][Olympic gold medal][Moved by the win]."

**ASSISTANT**
$[\{edge : [4659, 17264], influence\_value : 0.8\}, edge : [17264, 11799], influence\_value : 0.9\}]$

---

