# OpenReview forum: "Make Information Diffusion Explainable: LLM-based Causal Framework for Diffusion Prediction"
_NeurIPS.cc/2025/Conference — NeurIPS 2025 poster_

### Official Review · Reviewer_CJaf · 2025-06-25

**Clarity:** 3
**Significance:** 2
**Originality:** 2
**Rating:** 4
**Confidence:** 2

**Summary:**

This paper proposes an information diffusion prediction framework (MILD) based on large language models (LLMs). Unlike previous methods that focus only on social structure and propagation sequence, MILD leverages LLMs to infer the causal relationships of “who influences whom” within cascades. It comprehensively integrates multi-dimensional data such as social network structure, user activity, comment content, and temporal information to explicitly construct a directed diffusion influence graph, which is then incorporated into downstream diffusion prediction tasks.

**Questions:**

Can the authors further analyze or provide information on the inference efficiency and resource consumption of MILD when applied to larger-scale, longer cascades, or more complex social networks?

Is there any further external data or human-annotated experiments to validate the causal validity of the “who influences whom” chains inferred by the LLM?

Since the model utilizes sensitive user attributes and comment content, how can privacy and data compliance be ensured in practical applications? Is it possible to anonymize or obfuscate sensitive information?

**Ethical Concerns:**

["NO or VERY MINOR ethics concerns only"]

**Limitations:**

The authors have discussed some limitations, such as inference cost and the verifiability of LLM outputs, but the analysis of privacy and ethical risks is insufficient. It is recommended to include statements regarding data collection compliance and to further analyze the method's resource consumption and practical applicability in large-scale scenarios.

**Paper Formatting Concerns:**

The overall formatting is standard, with no obvious layout issues.

**Quality:**

3

**Strengths And Weaknesses:**

Strengths：

It skillfully integrates social relationships, temporal order, user profiles, and comment content, resulting in comprehensive and well-structured information utilization.

In addition to ablation studies, the paper also conducts empirical analyses from perspectives such as robustness and attention visualization.

Weaknesses：

MILD relies on large language model inference, which may incur excessive deployment and computational costs in ultra-large-scale social networks or real-time prediction scenarios, posing challenges for practical application.

Although LLMs can infer “who influences whom,” the inferred causal chains are difficult to fully verify in real-world scenarios and may introduce subjectivity or hallucinations.

---

> ### Author Rebuttal · Authors · 2025-07-31
>
> We sincerely appreciate your valuable feedback. We hope that the following clarifications help address your concerns.
>
> ### **[W1&Q1] Inference Efficiency**
>
> Thank you for your valuable feedback. We provide time complexity analysis and conduct new experiments evaluating the time cost of MILD. Results show that MILD remains SOTA effectiveness with low computational cost.
>
> **1. Theoretical Time Complexity**
>
> We provide a breakdown time complexity of each module as follows.
>
> |**Module**|**Time complexity**|
> |-|-|
> | Prompt Construction | $\mathcal{O}(\mid V_C\mid d_{max}^r+\mid V_C\mid\mid E_C\mid)$|
> | LLM Reasoning| $\mathcal{O}_{\text{LLM}}(L)$ |
> |Transformer Prediction| $\mathcal{O}_\text{Trans}(\mid V_C\mid)$ |
> |Overall MILD|$\mathcal{O}(\mid V_C\mid d_{max}^r+\mid V_C\mid\mid E_C\mid)+N\mathcal{O}_{\text{LLM}}(L)$|
>
> $|V_C|$ and $|E_C|$ denote the size of the cascade graph. $d_{\max}^r$ is the maximum out-degree in the cascade graph. $r$ is the graph density, which is a fixed hyperparameter. Thus, $O(\mid V_C\mid d_{max}^r+\mid V_C\mid \mid E_C\mid)$ is in polynomial time. For the LLM reasoning process, for each query, the complexity is determined by the token length of prompts $L$ and the LLM function itself $O_{\text{LLM}}(\cdot)$, which will decrease as prompts shorten and LLMs evolve. For Transformer-based prediction, the complexity is determined by the size of the cascade $|V_C|$ and the function $O_\text{Trans}(\cdot)$. In summary, as $O_{\text{Trans}}(\mid V_C\mid)\subseteq O_{\text{LLM}}(L)$, the time complexity of MILD is $O(|V_C| d_{max}^r+| V_C||E_C|)+N\ O_{\text{LLM}}(L)$. $N$ is the number of prompts, which is a controllable parameter.
>
> **2. Runtime Evaluation**
>
> **2.1 Efficiency on Larger-Scaled and More Complex Graphs**:
>
> We conducted new experiments to evaluate the time cost of MILD, showing that the overall runtime of MILD is acceptable and can be reduced easily by setting a small number of LLM reasoning prompts for various downstream requirements. Thus, MILD achieves an effective balance between computational efficiency and the gains in performance and explainability, thereby offering strong practical utility.
>
> |Datasets|#Node|Avg. Node Degree|Avg. Runtime per cascade|
> |:-|-|-|-|
> | News| 10,255| 16.90  | 39.77s |
> | Our Weibo | 31,061| 32.63  | 54.39s |
> | Original Weibo | 1,776,950  | 170.7  | 92.53s |
>
> In MILD, the LLM reasoning step introduces additional computational cost. However, we think this latency is acceptable for several reasons:
>
> - In practice, the number of prompts is controllable, which can be reduced for requirements in diverse scenarios.
> - The LLM reasoning process serves as a preliminary analysis step, and its results can be leveraged across various downstream applications (such as diffusion prediction and public opinion analysis). As it is performed before downstream tasks, it does not affect their real-time efficiency. Moreover, the rapid evolution of LLMs is continuously reducing inference times, which will further improve the efficiency of MILD.
> - Especially for diffusion prediction, the time cost of the LLM reasoning process is a worthwhile trade-off for the ~7% improvement in prediction accuracy and, crucially, the novel explainability our MILD framework provides. As we validated in Section 3.3, MILD is consistent with real public opinion.
>
>
> **2.2 Heuristic Algorithm Runtime on Larger-Scaled and More Complex Graphs:**
>
> We evaluate the time cost of our heuristic algorithm on a range of graphs with varying *scales* and *densities*, and *longer cascades*, demonstrating that our heuristic solution achieves strong efficiency. Specifically, it achieves 0.0334s per cascade, even for synthetic large-scale dense graphs with millions of nodes and an average degree of 120. On the Weibo platform with 1.7 million users, our heuristic solution can still achieve 0.1290s.
>
> | Datasets | News | Our Weibo | Original Weibo |
> | :- |-|-|-|
> | Avg. Runtime per cascade (s) | 0.0126s | 0.0198s| 0.1290s|
>
> **For different scaled graphs with different node degree:**
>
> | Degree 40 | Avg. Runtime per cascade (s) | Runtime Growth |
> |-|-|-|
> | Graph with 50000 nodes | 0.0050 |- |
> | Graph with 100000 nodes  | 0.0049 | ~1.0x|
> | Graph with 300000 nodes  | 0.0070 | ~1.4x|
> | Graph with 1000000 nodes | 0.0086 | ~1.7x|
>
> | Degree 80 | Avg. Runtime per cascade (s) | Runtime Growth |
> |-|-|-|
> | Graph with 50000 nodes | 0.0095 |- |
> | Graph with 100000 nodes  | 0.0100 | ~1.0x|
> | Graph with 300000 nodes  | 0.0115 | ~1.2x|
> | Graph with 1000000 nodes | 0.0124 | ~1.3x|
>
> | Degree 120 | Avg. Runtime per cascade (s) | Runtime Growth |
> |-|-|-|
> | Graph with 50000 nodes | 0.0139 | - |
> | Graph with 100000 nodes  | 0.0145 | ~1.0x|
> | Graph with 300000 nodes  | 0.0162 | ~1.2x|
> | Graph with 1000000 nodes | 0.0334 | ~2.4x|
>
> **2.3 Heuristic Algorithm Runtime on Longer Cascades:**
>
> | **Cascade Length** | **News Dataset** | **Weibo Dataset** |
> | :- | :- | :- |
> | 1-50 | 0.002s| 0.013s|
> | 50-100 | 0.002s| 0.013s|
> | 101-200| 0.012s| 0.020s|
> | 201-300| 0.029s| 0.044s|
> | 301-500| 0.217s| 0.212s|
>
>
> Overall, the above results indicate that MILD shows good efficiency across larger-scale and denser graphs, longer cascades, and more complex social networks.
>
> **3. Cost-Effective Deployment with Open-Source LLMs**
>
> We further conducted experiments with four open-sourced and two closed-source LLMs. Results show that open LLMs perform comparably to commercial ones, validating that MILD can be effectively deployed without reliance on costly commercial LLMs.
>
> | **News Dataset** | **H@10** | **H@50** | **H@100** | **M@10** | **M@50** | **M@100** |
> |-|-|-|-|-|-|-|
> | **GPT-4o (reported)** | 24.08| 31.31| 35.50| 15.49| 15.87| 15.92|
> | Gemini 2.5| 23.91| 31.13| 35.32| 15.38| 15.71| 15.82|
> | Claude 4-Sonnet| 23.97| 31.19| 35.42| 15.46| 15.78| 15.86|
> | LLaMA3-70B| 23.58| 30.28| 34.36| 15.21| 15.31| 15.34|
> | Qwen-turbo| 22.92| 30.06| 34.08| 14.74| 15.24| 15.29|
> | Qwen3 | 22.89| 29.87| 33.88| 14.69| 15.02| 15.16|
> | DeepSeek R1| 22.56| 29.91| 33.94| 14.68| 15.11| 15.17|
>
>
> | **Weibo Dataset**| **H@10** | **H@50** | **H@100** | **M@10** | **M@50** | **M@100** |
> |-|-|-|-|-|-|-|
> | **GPT-4o (reported)** | 14.73| 23.34| 28.18| 8.70| 9.09| 9.15 |
> | Gemini 2.5| 14.66| 23.33| 28.12| 8.64| 9.06| 9.16 |
> | Claude 4-Sonnet| 14.65| 23.31| 28.07| 8.63| 9.08| 9.10 |
> | LLaMA3-70B| 14.25| 22.92| 28.00| 8.40| 9.00| 9.06 |
> | Qwen-turbo| 14.20| 22.61| 27.93| 8.24| 8.66| 8.73 |
> | Qwen3 | 14.28| 22.73| 27.90| 8.13| 8.14| 8.23 |
> | DeepSeek R1| 14.31| 22.80| 27.94| 8.08| 8.11| 8.14 |
>
>
> **3.2 MILD Token Cost:**
>
> MILD is efficient and keeps token usage low, showing faster and cheaper use in real-world scenarios. Besides, MILD with open-source LLMs can achieve SOTA performance with 0 dollars.
>
> |Datasets|Avg. #Input tokens| **Avg. #Output tokens** |
> |-|-|-|
> | News  | 285.67 | 12.25|
> | Weibo | 647 | 24 |
>
> **Hard resources requirement:** In our evaluation, One Tesla V100 32G is enough.
>
> ---
>
> ### **[W2 & Q2] Validation of LLM-derived Chains**
>
> Thank you for emphasizing the validation of the LLM-derived $G_I$, which is the key in our MILD.
>
> In the submission paper's Sections 3.3, 4.1, and 4.2, we have conducted experiments with human-annotated and external data to verify "who influences whom" chains, validating the causal validity of this. Here, we clarify it more detailedly as follows.
>
> **[Sections 4.1 and 4.2]** Downstream Performance as the Primary Validation of $G_I$
>
> The most powerful evidence for the validity of the LLM-inferred influence graph ($G_I$) is its direct impact on the primary downstream task: **diffusion prediction**. Our framework significantly outperforms eight state-of-the-art baselines. A more accurate diffusion influence graph directly translates to more accurate predictions, and our SOTA results serve as strong empirical validation that $G_I$ is capturing meaningful, real diffusion dynamics that are overlooked by other models.
>
> **[Section 3.3]** Real-world Diffusion Pathways and Key Interactions as External Validation
>
> > **Real-world diffusion pathways as external data:**
>
> In Section 3.3 (E1), we carefully extracted actual diffusion chains on the real-world social platform as external data for evaluations. Specifically, real-world diffusion pathways always appear in reply comments, in the form of $\{A:\text{[comment]} @B:\text{[comment]}@C:\text{[comment]}\}$. Thus, we can extract these as the ground truth. Results (shown in Figure 4 in Section 3.3) indicate that our diffusion influence graph $G_I$ can truly capture these real-world information diffusion pathways.
>
> >  **Key interaction as external data:**
>
> In Section 3.3 (E2), we also manually extracted key interactions among users during the information diffusion process. Specifically, as the well-known communication theory, the Strength of Weak Ties, posits, \emph{"the strength of any tie is a function of the frequency and duration of interaction"}, we assessed whether $G_I$ observes participant interactions among participants within cascades. Key interactions are high-frequency interactions that happen in the real-world social platform.
>
> ---
>
> ### **[Q3] Data Privacy**
>
> We take data privacy and the ethical implications of our work with the utmost seriousness.
> _**All datasets are public research datasets**_ and have been processed with strict privacy-preserving measures. The prompts fed to the LLM contain only these anonymized or public IDs, timestamps, and comments, with no other Personally Identifiable Information (PII). Furthermore, the datasets used in our study have been **widely used by the research community**. Our use of this data is strictly for academic research purposes and complies with the terms of service under which the data was released. We will add a dedicated **"Ethics and Privacy Statement"** section to the paper to make these commitments explicit.
>
> ---
>
> Sincerely thank you again for the thoughtful and valuable comments.

---

### Official Review · Reviewer_iynN · 2025-06-30

**Clarity:** 2
**Significance:** 3
**Originality:** 3
**Rating:** 4
**Confidence:** 5

**Summary:**

This paper proposes a novel multi-step framework to predict information diffusion in social networks, namely, MILD (LLM-based Causal Framework for Diffusion Influence Derivation). One of the novelties of the proposed method is that inlike existing methods that are mainly based on GNNs and sequence models, it uses LLMs to explain underlying influence and infer a "who-influences-whom" diffusion graph. Experiments have been conducted to demonstrate that the proposed framework achieves SOTA accuracy as well as explainable representation of the diffusion mechanism.

**Questions:**

I am very curious to see authors' response to my comment that I posed earlier on the weaknesses of the method.

**Ethical Concerns:**

["NO or VERY MINOR ethics concerns only"]

**Final Justification:**

I have read authors responses and my concerns about time complexity and reproducibility have been addressed.

**Limitations:**

yes

**Quality:**

3

**Strengths And Weaknesses:**

Strengths:
1. Paper is well-written and easy to follow.
2. LLM used in the method helps with the explainability of the information diffusion process which is at the core of the paper's innovations.
3. The proposed method uses different types of inputs and information which make the method more reliable and stronger.
4. Also extensive experiments (including ablation studies) have been performed to support the proposed idea. Furthermore, experiments have been supported by rigorous theoretical discussions.
5. Extensive evaluations have been performed to show the superiority of the results and the proposed method in general.

Weaknesses:
1. Although this is a nice paper, I have two major concerns regarding (1) computational costs and (2) reproducibility of the results. First of all, the method is comprised of multiple steps that make the method very expensive and the process quite time-consuming. Second, because of LLM and numerous parameters, reproducibility is very challenging due to underlying LLM's reasoning abilities.

---

> ### Author Rebuttal · Authors · 2025-07-31
>
> We sincerely appreciate your constructive feedback. In response to the concerns regarding (1) computational costs and (2) reproducibility of LLMs, we add more experiments and theoretical analysis to mitigate your concerns.
>
> ### **[W1] Computational Costs**
>
> Thank you for this crucial question. We provide a theoretical analysis of time complexity and conduct new experiments evaluating the time cost of MILD. Results show that MILD remains SOTA effectiveness with low computational cost.  We will enhance the discussion in the final version and provide the experiments and analysis as follows:
>
> **1. Time Complexity**
>
> The time complexity of MILD mainly involves three modules, which can be conducted in polynomial time.
>
>
> |**Module**|**Time complexity**|
> |-|-|
> | Prompt Construction | $\mathcal{O}(\mid V_C\mid d_{max}^r+\mid V_C\mid\mid E_C\mid)$|
> | LLM Reasoning       | $\mathcal{O}_{\text{LLM}}(L)$   |
> |Transformer Prediction    | $\mathcal{O}_{\text{Trans}}(\mid V_C \mid)$ |
> |Overall MILD|$\mathcal{O}(\mid V_C\mid d_{max}^r+\mid V_C\mid\mid E_C\mid)+N\mathcal{O}_{\text{LLM}}(L)$|
>
> In our heuristic algorithm, $|V_C|$ and $|E_C|$ denote the size of the cascade graph. $d_{\max}^r$ is the maximum out-degree in the cascade graph. $r$ is the graph density, which is a fixed hyperparameter. Thus, $O(\mid V_C\mid d_{max}^r+\mid V_C\mid \mid E_C\mid)$ is in polynomial time. For the LLM reasoning process, for each query, the complexity is determined by the token length of prompts $L$ and the LLM function itself $O_{\text{LLM}}(\cdot)$, which will decrease as prompts shorten and LLMs evolve. For Transformer-based prediction, the complexity is determined by the size of the cascade $|V_C|$ and the function $O_\text{Trans}(\cdot)$. In summary, as $O_{\text{Trans}}(\mid V_C\mid)\subseteq O_{\text{LLM}}(L)$, the time complexity of MILD is $O(|V_C| d_{max}^r+| V_C||E_C|)+N\ O_{\text{LLM}}(L)$. $N$ is the number of prompts, which is a controllable parameter.
>
> **2. Runtime Evaluation**
>
> We further evaluate the efficiency and token costs as follows.
>
> **2.1 Average Runtime (in seconds per cascade):**
>
> | **Module** | **News Runtime**| **Weibo Runtime** |
> | :- | :- | :- |
> | Prompt Construction|0.012s|0.019s|
> | LLM Reasoning (per prompt)|1.53s|1.62s|
> | Transformer Prediction|0.38s|0.33s|
> | Total Time|39.77s|54.39s|
> | Avg. #Prompts|25.77|33.36|
>
> Both prompt construction and Transformer prediction are highly efficient, with average runtimes well under 1 second. While the LLM reasoning step introduces additional computational overhead, we think this latency is acceptable for several reasons:
>
> - The number of prompts is controllable, which can be reduced for requirements in diverse scenarios.
> - The LLM reasoning process serves as a preliminary analysis step, and its results can be leveraged across various downstream applications (such as diffusion prediction and public opinion analysis). As it is performed before downstream tasks, it does not affect their real-time efficiency. Moreover, the rapid evolution of LLMs is continuously reducing inference times, which will further improve the efficiency of MILD.
> - Especially for diffusion prediction, the time cost of the LLM reasoning process is a worthwhile trade-off for the ~7% improvement in prediction accuracy and, crucially, the novel explainability our MILD framework provides. As we validated in Section 3.3, MILD is consistent with real public opinion.
>
> **Hard resources requirement:** One Tesla V100 32GB is enough for our evaluation.
>
>
>
>
> -----
>
> ### **[W2] LLM Reproducibility**
>
> Thank you for highlighting the fundamental importance of reproducibility. In our evaluation, we have taken corresponding measures to ensure reproducibility.
>
> **1. _0-temperature_ Setting for Reproducibility**
> We fully agree with the reviewer's concern and have specifically addressed it in our work. As described in *Section 4, line 301*, we set the "temperature" hyperparameter to **0** to ensure reproducibility during all LLM reasoning steps.
>
> "Temperature" parameter directly controls the variability in LLM-generated responses. **As confirmed in the GPT-4 official technical report [1]**, temperature=0 ensures deterministic and stable outputs for the same input prompt. This practice is widely adopted in the research community to guarantee result consistency [2][3]. By following this standard, MILD can **guarantee the reproducibility** of the "who-influences-whom" diffusion graph.
>
> We have released the source code in the supplemental materials, and *we also promise to provide a verification code for LLM reproducibility.*
>
> [1] Gpt-4 technical report. arXiv preprint arXiv:2303.08774 (2023).
>
> [2] The effect of sampling temperature on problem solving in large language models. EMNLP (2024).
>
> [3] A survey of large language models. arXiv preprint arXiv:2303.18223 1.2 (2023).
>
>
>
> **2. Robustness across different LLMs**
>
> We evaluate MILD on six different state-of-the-art LLMs (four open-source models and two closed-source models). We will add these experiments to the final version.
>
> *Summary:* The results below show that MILD achieves stable performance across six LLMs. (1) MILD is broadly generalizable, working effectively with a range of open-source and closed-source LLMs. (2) The success of MILD is determined by the causal framework and prompting strategy, not relying on the reasoning ability of a single LLM, such as GPT-4o. (3) Open-source LLMs (e.g., LLaMA 3, Qwen Turbo, Deepseek R1) deliver comparable results, making MILD both widely accessible and cost-effective for the broader research community.
>
> **Performance with Various LLMs:**
>
> | **News Dataset**      | **H@10** | **H@50** | **H@100** | **M@10** | **M@50** | **M@100** |
> | --------------------- | -------- | -------- | --------- | -------- | -------- | --------- |
> | **GPT-4o (reported)** | 24.08    | 31.31    | 35.50     | 15.49    | 15.87    | 15.92     |
> | Gemini 2.5            | 23.91    | 31.13    | 35.32     | 15.38    | 15.71    | 15.82     |
> | Claude 4-Sonnet       | 23.97    | 31.19    | 35.42     | 15.46    | 15.78    | 15.86     |
> | LLaMA3-70B            | 23.38    | 30.28    | 34.36     | 15.01    | 15.31    | 15.34     |
> | Qwen-turbo            | 22.92    | 30.06    | 34.08     | 14.74    | 15.24    | 15.29     |
> | Qwen3                 | 22.89    | 29.87    | 33.88     | 14.69    | 15.02    | 15.16     |
> | DeepSeek R1           | 22.56    | 29.91    | 33.94     | 14.68    | 15.11    | 15.17     |
>
>
> | **Weibo Dataset**     | **H@10** | **H@50** | **H@100** | **M@10** | **M@50** | **M@100** |
> | --------------------- | -------- | -------- | --------- | -------- | -------- | --------- |
> | **GPT-4o (reported)** | 14.73    | 23.34    | 28.18     | 8.70     | 9.09     | 9.15      |
> | Gemini 2.5            | 14.66    | 23.33    | 28.12     | 8.64     | 9.06     | 9.16      |
> | Claude 4-Sonnet       | 14.65    | 23.31    | 28.07     | 8.63     | 9.08     | 9.10      |
> | LLaMA3-70B            | 14.25    | 22.92    | 28.00     | 8.40     | 9.00     | 9.06      |
> | Qwen-turbo            | 14.20    | 22.61    | 27.93     | 8.24     | 8.66     | 8.73      |
> | Qwen3                 | 14.28    | 22.73    | 27.90     | 8.13     | 8.14     | 8.23      |
> | DeepSeek R1           | 14.31    | 22.80    | 27.94     | 8.08     | 8.11     | 8.14      |
>
> ---
>
> We would like to improve the paper for all questions, and we hope these clarifications and additional results are helpful for your concerns.

---

### Official Review · Reviewer_2ybE · 2025-07-01

**Clarity:** 3
**Significance:** 2
**Originality:** 3
**Rating:** 4
**Confidence:** 4

**Summary:**

This paper proposes a causal framework based on LLMs called MILD to predict information diffusion on social platforms. The framework integrates four key factors (social connections, activity timelines, user profiles, and comments) to infer a diffusion influence graph, which explicitly illustrates the relationships and influences among users involved in information cascades. Unlike traditional models, MILD harnesses the strong reasoning capabilities of LLMs to provide a more accurate and explainable diffusion prediction structure. Experiments show that the graphs constructed by MILD improve the representation of real-world diffusion paths by 12% over traditional social network structures and outperform eight state-of-the-art methods in diffusion prediction performance by approximately 7%.

**Questions:**

Please refer to Strengths and Weaknesses.

**Ethical Concerns:**

["NO or VERY MINOR ethics concerns only"]

**Final Justification:**

The author has added a larger dataset and conducted more experiments on various LLMs, addressing concerns about insufficient dataset size and the generalizability of the method. However, I still have doubts regarding the efficiency of MILD because only MILD's efficiency data is provided, without any comparison to other methods. Therefore, I am unable to assess the relative efficiency of MILD. For this reason, I have decided to raise the score to Borderline accept.

**Limitations:**

No. In fact, Weibo is a large-scale dataset, but it is unclear whether the authors only obtained a subset or performed data reduction. If experiments could be conducted on the complete dataset, it could effectively address the limitations of this work.

**Quality:**

2

**Strengths And Weaknesses:**

## Strengths
1. Innovative causal framework. A novel framework that combines LLMs and multiple factors is proposed to address the issue that traditional methods cannot clearly identify key influential connections.
  2. Solid theoretical foundation. Provides detailed theoretical analysis and proofs, including complexity analysis of the problem and the theoretical basis for the solution.
  3. Strong empirical results. Experiments on real-world datasets demonstrate the effectiveness of the method, with performance significantly surpassing existing state-of-the-art approaches.
## Weaknesses
  1. Dataset limitations. The experiments mainly focus on two datasets, which may not fully reflect the characteristics of information diffusion on different types of social platforms. Could additional experiments on larger-scale datasets be provided? Furthermore, as far as I know, Weibo is a large-scale dataset with over 1.7 million users, as cited in reference [1]. Why are there only about 30,000 users in the dataset used by the authors? This suggests that the authors performed data reduction, but the paper does not explain this. It is recommended that the authors provide a detailed explanation of how and why the data was reduced.
  2. High computational complexity. The path search problem in the paper is proven to be NP-hard, and the heuristic-based solution may face computational challenges in large-scale applications. Could the authors provide a comparison of the time costs among various methods or showcase the time cost of MILD on larger-scale datasets?
  3. Heavy reliance on LLMs. The method is highly dependent on the reasoning ability of LLMs, and different LLMs may produce different results, affecting the stability of the approach. However, the paper only presents results based on GPT-4o and does not compare with other types or versions of LLMs, which may limit the generalizability and applicability of the results. It is suggested to add experimental results using more LLMs.

[1] Social influence locality for modeling retweeting behaviors. IJCAI 2013.

---

> ### Author Rebuttal · Authors · 2025-07-31
>
> We sincerely appreciate your valuable comments and hope the following responses help address your concerns.
>
> ---
>
> ### **[W1] Dataset Selection and Scalability**
>
> In the submission paper's Appendix D.1 (Lines 687-699), we have provided how and why we reduce the data. The size of reduced data is commonly adapted (~10k) [1,2,3]. Moreover, we have conducted new experiments on large networks (original Weibo) with 1.7M nodes as advised, reflecting that our approach achieves better performance against other methods.
>
> **1. Data Subsetting based on Localized Information Diffusion:**
>
> Our approach to subsetting the Weibo dataset, which is detailed in Appendix D.1 of our submission (Lines 687-699), is a standard and methodologically sound practice in network analysis [1,2,3]. The rationale is grounded in the principle that **information diffusion is typically a localized phenomenon**. A single cascade rarely engages all 1.7 million users on the platform; instead, it propagates within specific communities, often driven by Key Opinion Leaders (KOLs).
>
> Therefore, we first identify influential KOLs and then construct a representative subgraph of these KOLs and the users who actively engage with their content. This ensures our dataset captures a dense and relevant set of real-world diffusion cascades, which is a common practice for creating focused yet realistic benchmarks.
>
>   * As shown in Table 5 (Appendix D), the four commonly used benchmark datasets (without meta-text data) also have similar sizes (~10k).
>
> [1] Wang et al. Information Diffusion Prediction With Graph Neural Ordinary Differential Equation Network. In MM 2024.
>
> [2] Zhong et al. Information Diffusion Prediction via Cascade-Retrieved In-Context Learning. In SIGIR 2024.
>
> [3] Jiao et al. Enhancing Multi-Scale Diffusion Prediction via Sequential Hypergraphs and Adversarial Learning. In AAAI 2024.
>
> **2. Experiments on the Original 1.7M-Node Weibo Dataset**
>
> We conducted new experiments using the original Weibo graph with 1.7 million nodes. The results demonstrate that MILD remains both effective and efficient.
>
> *   **Effectiveness is Maintained:** We evaluated the same test set of cascades on both our subset and the full graph. The performance is nearly identical, confirming that our localized subgraph is highly representative and does not lead to a loss of predictive power.
>
> | **Graph Used for Evaluation**   | **H@50**   | **M@50**   |
> | :---------------------------------- | :--------- | :--------- |
> | Selected Weibo Subgraph (31k nodes) | **0.2337** | **0.0910** |
> | Original Weibo Graph (1.7M nodes)   | 0.2319 | 0.0903 |
>
> *   **Efficiency is Scalable:** The runtime of our path-finding algorithm, the most computationally intensive step, shows only a moderate increase, confirming MILD's practicality for large-scale networks.
>
> | **Dataset**| **# Nodes** | **Avg. Node Degree** | **Avg. Runtime per Cascade** | **Runtime Growth** |
> |:-| :---------- | :------------------- | :--------------------------- | :----------------- |
> | Selected Weibo | 31,061  | 32.63| 0.0198s | 1x|
> | Original Weibo | 1,776,950   | 170.7| 0.1290s | ~6.5x  |
>
> **3. Data Diversity:**
>
> We sincerely appreciate your thoughtful feedback. As discussed in the "Limitations" Section in our paper, the availability of such datasets is limited, composed of complete user profiles, social relationships, diffusion dynamics, and users' comments. As far as we know, Weibo is a popular Chinese social media platform that is well-representative. We are actively seeking additional datasets and will gladly incorporate them in future work. We would appreciate it if you could provide more data resources or links to enhance our work.
>
> ---
>
>
> ### **[W2] Computational Complexity**
>
> We provide a detailed time complexity analysis of MILD and new experiments evaluating the practical time cost of heuristic-based solutions and MILD, demonstrating that our heuristic algorithm achieves high efficiency on million-level large dense graphs.
>
> **1. Theoretical Time Complexity Analysis:**
> As proven in Section 3.2, the time complexity of our heuristic-based solution is $\mathcal{O}(|V_C|d_{\max}^r + |V_C||E_C|)$, which is not directly determined by the scale of the entire social network. $|V_C|$ and $|E_C|$ are the number of nodes and edges in the cascade graph $G_C$. $d_{\max}^r$ is the maximum out-degree in the cascade graph. $r$ is the graph density, which is a fixed hyperparameter. Thus, $O(\mid V_C\mid d_{max}^r+\mid V_C\mid\mid E_C\mid)$ is in polynomial time. For the whole MILD framework, the time complexity is $O(|V_C| d_{max}^r+|V_C||E_C|)+N\ O_{\text{LLM}}(L)$, where $L$ is the token length of prompts. $N$ is the number of prompts, which is a controllable parameter we set. Thus, the complexity of MILD is minimally affected by the overall size of the social network, further supporting the efficiency of our approach in large‐scale graphs.
>
>
>
>
> **2. Time Cost of MILD on Larger-Scale Datasets:**
> We added new experiments evaluating our heuristic solution on a range of graphs with varying scales and graph densities. Moreover, we conducted experiments evaluating the overall runtime of MILD on larger and more complex social networks (original Weibo).
>
> Remarkably, even for large-scale dense graphs with millions of nodes and an average degree of 120, our heuristic solution achieves millisecond-level efficiency (0.0334s per cascade). On the real-world Weibo dataset with 1.7 million users, our heuristic solution can still achieve 0.1290s, further supporting the significance of our contribution.
>
> The overall runtime of MILD is acceptable and can be reduced easily by setting a small number of LLM reasoning prompts for various downstream requirements.
>
> **2.1 Runtime of Heuristic Algorithm per Cascade:**
>
> | Datasets   | #Node | Avg. Node Degree | Avg. Runtime per cascade |
> |:-|-|-|-|
> | News   | 10,255| 16.90| 0.0126s |
> | Our Weibo  | 31,061| 32.63| 0.0198s |
> | Original Weibo | 1,776,950 | 170.7| 0.1290s |
>
>
> For different scaled graphs with average node degree 40:
>
> | **Graph Scale**  | **Avg. Runtime per cascade (s)** | **Runtime Growth** |
> |-|-|-|
> | Graph with 50000 nodes   | 0.0050  | - |
> | Graph with 100000 nodes  | 0.0049  | ~1.0x  |
> | Graph with 300000 nodes  | 0.0070  | ~1.4x  |
> | Graph with 1000000 nodes | 0.0086  | ~1.7x  |
>
> For different scaled graphs with average node degree 120:
>
> | **Graph Scale**  | **Avg. Runtime per cascade (s)** | **Runtime Growth** |
> |-|-|-|
> | Graph with 50000 nodes   | 0.0139  | - |
> | Graph with 100000 nodes  | 0.0145  | ~1.0x  |
> | Graph with 300000 nodes  | 0.0162  | ~1.2x  |
> | Graph with 1000000 nodes | 0.0334  | ~2.4x  |
>
> **2.2 Total Time Cost of MILD per Cascade**:
>
> | **Datasets**   | **#Node** | **Avg. Runtime per cascade (s)** |
> |:-| ---------- |-|
> | News   | 10,255 | 39.77s  |
> | Our Weibo  | 31,061 | 54.39s  |
> | Original Weibo | 1,776,950  | 92.53s  |
>
> In MILD, the LLM reasoning step introduces additional computational cost. However, we believe this latency is acceptable for several reasons:
>
> - The LLM inference is a one-time, preliminary analysis whose results can be cached and reused for multiple downstream tasks (e.g., diffusion prediction, popularity prediction). As it is performed before downstream tasks, it does not affect their real-time efficiency. Moreover, the rapid evolution of LLMs is continuously reducing inference times, which will further improve the efficiency of MILD.
>
> - Especially for diffusion prediction, the time cost of the LLM reasoning process is a worthwhile trade-off for the ~7% improvement in prediction accuracy and, crucially, the novel explainability our MILD framework provides. As we validated in Section 3.3, MILD is consistent with real public opinion.
>
> Thus, MILD achieves an effective balance between computational efficiency and the gains in performance and explainability, thereby offering strong practical utility.
>
> ---
>
> ### **[W3] Generalization of MILD on Diverse LLMs**
>
> We thank the reviewer for highlighting this key consideration, which is indeed crucial for the robustness of any LLM-based approach. We conducted extensive new experiments using **six different prominent LLMs**, covering both advanced open-source and closed-source models. *Results show remarkable consistency across different LLMs on most metrics.*
>
> **Performance of MILD with Different LLMs:**
>
> | **News Dataset**  | **H@10** | **H@50** | **H@100** | **M@10** | **M@50** | **M@100** |
> |-|-|-|-|-|-|-|
> | **GPT-4o (reported)** | 24.08| 31.31| 35.50 | 15.49| 15.87| 15.92 |
> | Gemini 2.5| 23.91| 31.13| 35.32 | 15.38| 15.71| 15.82 |
> | Claude 4-Sonnet   | 23.97| 31.19| 35.42 | 15.46| 15.78| 15.86 |
> | LLaMA3-70B| 23.58| 30.28| 34.36 | 15.21| 15.31| 15.34 |
> | Qwen-turbo| 22.92| 30.06| 34.08 | 14.74| 15.24| 15.29 |
> | Qwen3| 22.89| 29.87| 33.88 | 14.69| 15.02| 15.16 |
> | DeepSeek R1   | 22.56| 29.91| 33.94 | 14.68| 15.11| 15.17 |
>
>
> | **Weibo Dataset** | **H@10** | **H@50** | **H@100** | **M@10** | **M@50** | **M@100** |
> |-|-|-|-|-|-|-|
> | **GPT-4o (reported)** | 14.73| 23.34| 28.18 | 8.70 | 9.09 | 9.15  |
> | Gemini 2.5| 14.66| 23.33| 28.12 | 8.64 | 9.06 | 9.16  |
> | Claude 4-Sonnet   | 14.65| 23.31| 28.07 | 8.63 | 9.08 | 9.10  |
> | LLaMA3-70B| 14.25| 22.92| 28.00 | 8.40 | 9.00 | 9.06  |
> | Qwen-turbo| 14.20| 22.61| 27.93 | 8.24 | 8.66 | 8.73  |
> | Qwen3| 14.28| 22.73| 27.90 | 8.13 | 8.14 | 8.23  |
> | DeepSeek R1   | 14.31| 22.80| 27.94 | 8.08 | 8.11 | 8.14  |
>
>
> ---
>
> We sincerely hope that these clarifications are helpful, and we will improve further on all the mentioned questions.

---

> ### Comment · Reviewer_2ybE · 2025-08-06
>
> Thank you for adding a larger dataset and conducting more experiments on various LLMs, which addresses concerns about insufficient dataset size and the generalizability of the method. However, I still have doubts about the efficiency of MILD, as only the efficiency data for MILD is provided without any comparison to other methods. Therefore, I am unable to assess the relative efficiency of MILD. For these reasons, I have decided to adjust the score to Borderline accept.

---

> > ### Author Response · Authors · 2025-08-06
> >
> > We sincerely thank you for this insightful comment. We agree completely and have performed additional experiments to provide a comprehensive view.
> >
> > ---
> > ---
> >
> > ### **1. Efficiency Benchmarking**
> >
> > We first benchmark the inference efficiency of MILD’s prediction-only module against state-of-the-art prediction models. The tables below present the inference runtime and GPU memory usage.
> >
> > **Table 1: Inference Runtime** (per batch of 64 cascades)
> >
> > | Methods | News Dataset | Weibo Dataset |
> > | :-- | :--: | :---: |
> > | MINDS | 1.66s|1.95s|
> > | MGCL  | 0.59s|0.62s|
> > | GODEN | 0.94s|1.32s|
> > | CARE  | 1.18s|1.44s|
> > | **MILD (ours)** |  **0.38s**   |   **0.33s**   |
> >
> >
> > **Table 2: GPU Memory Usage (GB)**
> >
> > | Methods | News Dataset | Weibo Dataset |
> > | :-- | :--: | :---: |
> > | MINDS |10.51 |18.87 |
> > | MGCL  |4.86 | 4.94 |
> > | GODEN |7.73 | 12.09 |
> > | CARE  |8.42 | 12.76 |
> > | **MILD (ours)** | **2.88** | **3.03** |
> >
> > The results show that MILD’s prediction-only module achieves **the best efficiency and requires less GPU memory**. This is a direct result of our design: MILD employs a subgraph-based framework where the computational complexity is confined **within the small cascade subgraph**. In contrast, other models must process much larger graphs (e.g., the entire social network) during inference, leading to higher overhead.
> >
> > ---
> >
> > ### **2. Clarification on the LLM's Role: Offline Analysis vs. Online Prediction**
> >
> > We hope to clarify the LLM roles between *offline analysis* and *online prediction module* of MILD.
> >
> > The LLM-powered causal discovery is used exclusively as a **one-time, offline preprocessing step**. In this paper, the online diffusion prediction is a downstream task for evaluating the LLM-derived quality.
> >
> > -   The efficiency experiments presented above in Tables 1 and 2 pertain only to the prediction module, ensuring a fair and direct comparison with other prediction models.
> > -   The purpose of the LLM-derived diffusion analysis is to generate a high-quality influence graph ($Z$) by interpreting user interactions and content semantics. This causal analysis is *agnostic to specific downstream tasks*. Therefore, it is designed as a **one-time, offline preprocessing step.**
> > -   This design intentionally separates the reasoning of LLMs from the fast prediction task. Once the influence graph $Z$ is generated, it is used by the efficient prediction module or any other tasks without requiring any further LLM inference.
> > -   The offline-derived graph $Z$ also enables **other tasks for deeper causal analysis**, as shown in our paper (Sec. 3.3), where it is used to identify key interactions (E1), critical diffusion pathways (E2), and denoise the social graph (E3&E4).
> >
> > ---
> >
> > ### **3. Generalizability and Value of the LLM-Derived Influence Graph ($Z$)**
> >
> > We further show that the LLM-derived influence graph ($Z$) is a **powerful, generalizable asset**. We enhance several baselines by incorporating LLM-derived graph $Z$ as an additional input.
> >
> > Below, we select **attention-based models**, which are DyHGCN, MSHGAT, MGCL, and GODEN. We then inject the LLM-derived influence graph $Z$ into their attention module, mirroring our design in Eq. (10). The notation "+ $Z$" indicates the model is using our LLM-derived influence graph.
> >
> > **Table 3: Prediction Performance Boost on the News Dataset with Graph $Z$**
> >
> > | Methods |Hits@100 | MAP@100 |
> > | :-- | :---: | :---: |
> > | DyHGCN|  27.31  |  10.80  |
> > | DyHGCN + $Z$ | **29.55 (+2.24)** | **11.74 (+0.94)** |
> > | MSHGAT|  28.85  |  11.99  |
> > | MSHGAT + $Z$ | **30.56 (+1.71)** | **12.71 (+0.72)** |
> > | MGCL  |  31.93  |  13.24  |
> > | MGCL + $Z$ | **33.57 (+1.64)** | **14.12 (+0.88)** |
> > | GODEN |  33.17  |  13.73  |
> > | GODEN + $Z$| **34.61 (+1.44)** | **14.78 (+1.05)** |
> > | **MILD (ours)** |**35.52**|**15.91**|
> >
> >
> > **Table 4: Prediction Performance Boost on the Weibo Dataset with Graph $Z$**
> >
> > | Methods |Hits@100 |MAP@100 |
> > | :-- | :---: | :--: |
> > | DyHGCN|  18.50  |  6.27  |
> > | DyHGCN + $Z$ | **19.73 (+1.23)** | **6.91 (+0.64)** |
> > | MSHGAT|  21.77  |  6.53  |
> > | MSHGAT + $Z$ | **22.76 (+0.99)** | **7.26 (+0.73)** |
> > | MGCL  |  24.75  |  8.03  |
> > | MGCL + $Z$ | **25.53 (+0.78)** | **8.49 (+0.46)** |
> > | GODEN |  26.71  |  8.52  |
> > | GODEN + $Z$| **27.53 (+0.82)** | **8.93 (+0.41)** |
> > | **MILD (ours)** |**28.17**|**9.17**|
> >
> > We hope these results can validate that the causal influence graph is a **modular and generalizable technique**, which can be well transferred to other baselines and thus improve their prediction.
> >
> > ---
> >
> > Hope these additional results and clarifications can address your concerns. We will add these results to our paper, which will truly improve the paper's quality.
> >
> > Sincerely thank you again for your positive comments and score adjustment!

---

> > ### Author Response · Authors · 2025-08-08
> >
> > Dear Reviewer,
> >
> > We sincerely thank you for recognizing our additional experiments on larger datasets and various LLMs. We will add these experiments to our paper, which will truly improve our paper quality.
> >
> > As a follow-up, we have conducted further *efficiency comparisons with other prediction models* and provided *more in-depth analysis*. These results have been added to our additional comments, and we hope they are useful for your earlier concerns.
> >
> > As the discussion deadline approaches, please let us know if any further clarification would be helpful for your assessment. We would be very glad to provide and discuss it.
> >
> > We greatly appreciate and value your constructive feedback, and will make careful revisions for these.
> >
> > Best regards,
> >
> > Authors of Submission 6744

---

### Official Review · Reviewer_FSjJ · 2025-07-01

**Clarity:** 3
**Significance:** 3
**Originality:** 3
**Rating:** 5
**Confidence:** 4

**Summary:**

This paper addresses the problem of information diffusion prediction. Noting the limitations of prior work based on graph neural networks and sequence models—particularly their lack of interpretability—the authors propose a Large Language Model (LLM)-based causal framework for diffusion influence derivation (MILD). The framework integrates four key factors underlying social diffusion dynamics. Experimental results demonstrate that the proposed approach outperforms traditional methods in both predictive performance and interpretability.

**Questions:**

About LLMs: Can the proposed method be adapted to use other open-source LLMs? Additionally, does the prompt design require few-shot examples to achieve good performance?

About the causal mechanism: Beyond inspiring the use of LLMs to derive the diffusion influence graph, does the causal framework serve any additional role in the model? How is the do-operator concretely instantiated within the model architecture or inference process?

On Equation 10: Why do the authors choose to inject the diffusion influence using a +\alpha A formulation instead of a masked A formulation? What are the advantages or motivations behind this design choice?

**Ethical Concerns:**

["NO or VERY MINOR ethics concerns only"]

**Final Justification:**

I am positive about this work from the very beginning. The clarification made by the author further justifies my decision. I keep my score.

**Limitations:**

Yes

**Paper Formatting Concerns:**

1.	Line 218， Figure 5 (a) should be Figure 4(a)
2.	Line 228,  Figure 5(b) should be Figure 5(b)

**Quality:**

4

**Strengths And Weaknesses:**

Strengths:
1.	The motivation of the paper is sound. Building on the observation that traditional models lack interpretability, the authors propose a causal inference framework enhanced by large language models (LLMs), which is a well-founded approach.
2.	The paper is well-written and easy to follow.
3.	The authors conduct extensive experimental analysis, including comparisons with a wide range of baselines, ablation studies, and empirical evaluations, making the results relatively solid.
4.	The proposed MILD model demonstrates clear improvements over many baseline methods.

Weaknesses:
1.	The core idea of the paper is essentially to leverage LLMs to capture a diffusion influence graph, which is then used for downstream prediction. However, I remain unconvinced about the effectiveness or necessity of the causal inference aspect as claimed. (See Questions)
2.	The discussion around the use of LLMs is somewhat insufficient. It would be beneficial to explore different LLM variants (e.g., open-source models), and to elaborate on the prompt design—such as whether few-shot prompting is necessary.
3.	The paper lacks analysis of important hyperparameters, such as the influence of alpha.

---

> ### Author Rebuttal · Authors · 2025-07-31
>
> We are deeply grateful for the positive comments and insightful feedback. We hope that the following clarifications are helpful for your concerns.
>
> ---
>
> ### **[W1&Q2] Role of Causal Framework**
>
> The causal framework is integral to our model's design. We clarify the additional role of the causal framework in MILD and the concrete do-operation in the model architecture.
>
> **1. MILD Under Causal Framework:**
>
> MILD is strictly designed to satisfy three front-door criteria in the causal framework to make a more accurate diffusion prediction. (C1) $Z$ intercepts all directed paths from $C$ to $Y$. When $Z$ can represent the information diffusion itself, $Z$ can intercept all directed paths from $C$ to $Y$. To achieve it, MILD captures almost all necessary and sufficient diffusion information to derive $Z$ based on communication theories to make sure that $Z$ can represent the nature of the information cascade. (C2) there is no unblocked back-door path from $C$ to $Z$, and (C3) all backdoor paths from $Z$ to $Y$ are blocked by $C$. MILD extract observed information cascades from the real-world social platform as $C$ to block the backdoor path from $Z$ to $Y$, which is $Z\leftarrow C\leftarrow U\rightarrow Y$, satisfying (C2) and (C3).
>
>
> **2. Do-operation analysis:**
>
> $P(Y|do(C))$ contains two do-operations, which can be formulated as $P(Y|do(C))=\sum\limits_{z\in Z} P(Y|do(Z))P(Z|do(C))$ based on the front-door criterion (C1). In MILD, $do(C)$ is to select several users who can form an observed information cascade, forcing $C=c$. $do(Z)$ operation is to derive diffusion influence representation $z$ based on given interventions, which are designed potential diffusion paths $P_C$, temporal order $T$, user influence and activity levels $O$, and comments $B$ from specific cascade $c$, forcing LLM to derive $Z=z$. In the $Z$ space, given the observed cascade $c$ without interventions, $Z$ contains all possible random diffusion processes. Thus, $P(Z|do(C))$ can be assumed as $P(z|do(C))=\frac{1}{|Z|}, z\in Z$. With interventions of $Z$, $P(Y|do(Z))$ is the probability of Transformer prediction $Y$ conditioned on our derived diffusion influence graph $z=G_I$.
>
> Additionally, we also conducted an ablation study (in Table 3) of the comparison of our $do(Z)$ operation and other random $z\in Z$. We replaced our $do(Z)$ operation with random $z\in Z$ (*w/o Influence bias*) and with social graph $U$ (*w Social bias*), leading to a significant drop in performance, confirming the critical role of $do(Z)$ operation and our causal framework. More detailed analysis can be found in _Appendix C.1 and C.2_.
>
> ---
>
> ### **[W2 & Q1] Adaptability to LLM Variants and Prompting Strategy**
>
> Thank you for your professional suggestions! Following your advice, we conducted additional experiments on LLM variants and few-shot prompting, and we will include these results in the final manuscript.
>
> **1. Performance on Diverse LLMs:**
>
> MILD can be effectively adapted to a wide range of LLMs. We evaluated its performance using six different models, including four open-source alternatives. The results below show that the performance is robust across all variants.
>
> | **News Dataset**      | **H@10** | **H@50** | **H@100** | **M@10** | **M@50** | **M@100** |
> | --------------------- | -------- | -------- | --------- | -------- | -------- | --------- |
> | **GPT-4o (reported)** | 24.08    | 31.31    | 35.50     | 15.49    | 15.87    | 15.92     |
> | Gemini 2.5            | 23.91    | 31.13    | 35.32     | 15.38    | 15.71    | 15.82     |
> | Claude 4-Sonnet       | 23.97    | 31.19    | 35.42     | 15.46    | 15.78    | 15.86     |
> | LLaMA3-70B            | 23.58    | 30.28    | 34.36     | 15.21    | 15.31    | 15.34     |
> | Qwen-turbo            | 22.92    | 30.06    | 34.08     | 14.74    | 15.24    | 15.29     |
> | Qwen3                 | 22.89    | 29.87    | 33.88     | 14.69    | 15.02    | 15.16     |
> | DeepSeek R1           | 22.56    | 29.91    | 33.94     | 14.68    | 15.11    | 15.17     |
>
>
> | **Weibo Dataset**     | **H@10** | **H@50** | **H@100** | **M@10** | **M@50** | **M@100** |
> | --------------------- | -------- | -------- | --------- | -------- | -------- | --------- |
> | **GPT-4o (reported)** | 14.73    | 23.34    | 28.18     | 8.70     | 9.09     | 9.15      |
> | Gemini 2.5            | 14.66    | 23.33    | 28.12     | 8.64     | 9.06     | 9.16      |
> | Claude 4-Sonnet       | 14.65    | 23.31    | 28.07     | 8.63     | 9.08     | 9.10      |
> | LLaMA3-70B            | 14.25    | 22.92    | 28.00     | 8.40     | 9.00     | 9.06      |
> | Qwen-turbo            | 14.20    | 22.61    | 27.93     | 8.24     | 8.66     | 8.73      |
> | Qwen3                 | 14.28    | 22.73    | 27.90     | 8.13     | 8.14     | 8.23      |
> | DeepSeek R1           | 14.31    | 22.80    | 27.94     | 8.08     | 8.11     | 8.14      |
>
>
>
> **2. Few-Shot Prompting:**
>
> We further conducted experiments of few-shot prompt engineering and found that **few-shot prompting is an effective prompt engineering technique, but may not be necessary for our MILD**.
>
> We evaluate 0-shot, 3-shot, and 5-shot prompting on the News Dataset as follows:
>
> | **Prompt Type** | **H@50** | **M@50** |
> | :-------------- | :------- | :------- |
> | 0-shot          | 0.3133   | 0.1585   |
> | 3-shot          | 0.3029   | 0.1501   |
> | 5-shot          | 0.3016   | 0.1467   |
>
> We observed a moderate performance decrease with few-shot examples for two primary reasons:
>
> *   **Confused by Noisy Patterns:** The complexity of real-world information diffusion cannot be easily captured in a few examples. Providing shots may bias the LLM toward simplified or incorrect patterns.
> *   **Increased Hallucination:** We found that with few-shot examples, the LLM was more likely to hallucinate user IDs or relationships from the examples rather than reasoning based on the provided cascade data.
>
> Given these findings, our zero-shot prompt, which is carefully designed based on communication theories, proves to be the most robust and effective approach.
>
> ---
>
> ### **[W3 & Q3] Hyperparameter $\alpha$ and Injection Formulation**
>
> Thank you for your careful reading and the insightful question about the injection formulation in Eq.(10) in Section 3.4.
>
> **1. Hyperparameter Analysis for $\alpha$:**
> We recognize the importance of this hyperparameter. In **Appendix A.1 (Figure 9)**, we have provided a sensitivity analysis for $\alpha$. The results show the model's performance across a reasonable range of $\alpha$ values (0.5 to 1.5).
>
> **2. Rationale for $+\alpha A$ Formulation:**
> We chose to inject the influence graph $A_I$ via an additive combination ($(X + \alpha A_I)H$) rather than a hard mask for two primary reasons:
>
> *   *Preserving Implicit Correlation:* A mask would completely discard the implicit correlation (e.g., shared interests) for user pairs not present in $A_I$. Our additive approach allows the model to leverage both the explicit LLM-inferred influence paths and the implicit user correlation, treating them as complementary sources of information.
>     - Example: Two people are not causally influenced, but liking the same content implies that they have the same interests (such implicit correlations are also useful for prediction).
> *   *Flexibility and Gradient Flow:* The $+\alpha A$ formulation creates a "soft" augmentation, allowing the model to learn the relative importance of the two different semantics. This provides a smoother optimization landscape compared to the hard constraints imposed by masking.
>
> **3. Empirical Comparison with Masking:**
>
> Masking operation is a reasonable observation. We provide an experiment to compare adding with a masking strategy, and we will add this experiment to the final version.
>
> | **Strategy**           | **H@50 (News)** | **M@50 (News)** | **H@50 (Weibo)** | **M@50 (Weibo)** |
> | :--------------------- | :-------------- | :-------------- | :-------------- | :-------------- |
> | Masked $A$             | 29.02           | 13.75           | 20.53           | 7.96            |
> | **+$\alpha A$ (Ours)** | **31.33**       | **15.85**       | **23.37**       | **9.10**        |
>
> The performance drop with masking suggests that even if some user-pairs do not represent the primary causal path for a given cascade, it may still contain useful relational information (e.g., shared interests). Discarding this information entirely is detrimental.
>
> ---
>
>
> **[Typos]** Thank you for your careful proofreading. We will correct the figure references in Lines 218 and 228 and conduct a more thorough proofread of the revised manuscript.
>
> ---
>
> Sincerely thanks for the encouraging and valuable feedback, and we hope these clarifications are helpful for addressing your concerns :)

---

> > ### Comment · Reviewer_FSjJ · 2025-08-05
> >
> > I feel that my concerns are sufficiently addressed. I choose to keep my score. I will keep an eye on the discussions with other reviewers.

---

> > > ### Author Response · Authors · 2025-08-05
> > >
> > > Thank you sincerely for your positive comments and feedback!
> > >
> > > We are glad that our response is useful. We will further improve the quality of our paper accordingly based on your constructive comments.

---

### Note · Authors · 2025-08-12

Dear Chairs,

We sincerely thank all reviewers for their constructive and valuable feedback. We are also grateful for the uniformly positive ratings, which acknowledge the main contributions of our work, MILD, as summarized below:

- **Innovative causal framework** (Reviewers FSjJ, 2ybE, iynN, CJaf)
- **Enhanced explainability** (Reviewers iynN, FSjJ)
- **Solid theoretical foundation** (Reviewers FSjJ, 2ybE, iynN)
- **Extensive experimental evaluation** (Reviewers FSjJ, 2ybE, iynN, CJaf)
- **Strong empirical results** (Reviewers FSjJ, 2ybE, iynN, CJaf)

Based on their comments, we identified two primary concerns: **(1) runtime efficiency** and **(2) generalization across different LLMs**.
We highly value these suggestions and mainly addressed both with additional experiments, as outlined below.

- **Efficiency:** We conducted additional experiments on **large-scale graphs**, **complex graphs with varying densities**, and **long diffusion cascades** (in response to Reviewer 2ybE [W1, W2], Reviewer iynN [W1], and Reviewer CJaf [W1, Q1]), validating MILD’s strong runtime efficiency. We also provided **efficiency benchmarking** against other prediction models (to Reviewer 2ybE) to directly assess MILD’s competitiveness. Results show that:
  - MILD’s predictor achieves **better efficiency and GPU memory usage** among all competitors. This advantage stems from MILD’s subgraph-based framework within *small cascade subgraphs*.
  - The LLM-powered causal discovery is performed only once as an offline pre-analysis step, incurring no runtime cost during prediction. The LLM-derived causal influence graph is both **plug-and-play and generalizable**, enabling straightforward integration into other baselines, and supporting additional tasks for deeper causal analysis (as evaluated in Sec 3.3).

- **Generalization:** We evaluated MILD on six different LLMs, including *four closed-source and two open-source* (in response to Reviewer FSjJ [W2, Q1], Reviewer 2ybE [W3], Reviewer iynN [W2], and Reviewer CJaf [W1, Q1]). Across all models and diverse prompt designs, **MILD consistently shows robust generalization and maintains state-of-the-art performance**.

We will carefully add these additional results into the final paper, along with deeper explanations. We once again extend our sincere gratitude to the Chairs and all reviewers for their time and dedication.

Best regards,

Authors of Submission 6744

---

### Decision · Program_Chairs · 2025-09-17

**Decision:**

Accept (poster)

**Comment:**

The paper introduces an LLM-based causal framework to predict information diffusion on social platforms. All the reviewers are mildly positive or positive about the paper after rebuttal.